# 🦍APE: FASTER AND LONGER CONTEXT-AUGMENTED GENERATION VIA ADAPTIVE PARALLEL ENCODING

**Xinyu Yang**
CMU
xinyuagi@cmu.edu

**Tianqi Chen**
CMU, NVIDIA
tqchen@cmu.edu

**Beidi Chen**
CMU
beidic@cmu.edu

## ABSTRACT

Context-augmented generation (CAG) techniques, including RAG and ICL, require the efficient combination of multiple contexts to generate responses to user queries. Directly inputting these contexts as a sequence introduces a considerable computational burden by re-encoding the combined selection of contexts for every request. To address this, we explore the promising potential of parallel encoding to independently pre-compute and cache each context's KV states. This approach enables the direct loading of cached states during inference while accommodating more contexts through position reuse across contexts. However, due to misalignments in attention distribution, directly applying parallel encoding results in a significant performance drop. To enable effective and efficient CAG, we propose Adaptive Parallel Encoding (**APE**), which brings shared prefix, attention temperature, and scaling factor to align the distribution of parallel encoding with sequential encoding. Results on RAG and ICL tasks demonstrate that APE can preserve 98% and 93% sequential encoding performance using the same inputs while outperforming parallel encoding by 3.6% and 7.9%, respectively. It also scales to many-shot CAG, effectively encoding hundreds of contexts in parallel. Efficiency evaluation shows that APE can achieve an end-to-end $4.5\times$ speedup by reducing $28\times$ prefilling time for a 128K-length context. The code is available at https://github.com/Infini-AI-Lab/APE.

## 1 INTRODUCTION

Recent advances in context-augmented generation (CAG), particularly retrieval-augmented generation (RAG) (Gupta et al., 2024; Gao et al., 2023) and in-context learning (ICL) (Dong et al., 2022; Wei et al., 2022), have been widely adopted in large language models (LLMs) (Dubey et al., 2024; Achiam et al., 2023), improving their ability to generalize to unseen tasks with contextual information, as demonstrated in Figure 1 (top). These techniques employ a *sequential encoding* process to ground LLM inputs with knowledge from external sources: concatenating the retrieved texts into one sequence, and encoding the sequence into key-value (KV) states as the context for subsequent queries. While this new, significantly longer input improves performance, the increased latency in context prefilling becomes a bottleneck in tasks that require long inputs but generate short outputs (Bai et al., 2023; Agarwal et al., 2024; Jiang et al., 2024b). For example, prefilling a 128K context can take 17 seconds, whereas generating 256 tokens requires only 6 seconds. This discrepancy leaves significant room to improve the practical efficiency of CAG systems when deployed in real-world applications (Liu, 2022; Chase, 2022).

Since these contexts are typically predetermined and stored independently in external databases (Zayarni et al., 2024; Douze et al., 2024), naively/bruteforcely one might accelerate CAG systems by pre-caching them for direct loading during inference. However, for autoregressive LLMs, the KV states are inherently context-dependent, which means that the KV states for the same text vary based on the preceding context. This dependency necessitates caching all possible context permutations, leading to factorial growth in memory requirements as the database size increases. For instance, caching all permutations of just ten 256-token texts for the LLAMA-3-8B model would demand an impractical 22 PB of memory.

To address this issue, *parallel encoding* (Ratner et al., 2022; Yen et al., 2024; Li et al., 2024; Sun et al., 2024) encodes each context into KV states separately, ensuring that tokens from different contexts cannot be attended in the encoding phase. Next, the on-the-fly generation is started by prefilling user queries, which can attend to the cached KV states from all contexts without re-encoding, offering two benefits:

**Pre-caching Contexts for Fast Inference:** Contexts from the external sources can be pre-computed and cached into KV states for direct loading during inference. Additionally, this approach allows for cost-free manipulation of contexts, including insertion, deletion, replacement, and swapping operations.

**Re-using Positions for Long Context:** Contexts can be inserted into the same range of positions in an LLM's context window, allowing for more and longer context chunks. This also avoids the "positional bias" in context ordering (Liu et al., 2024a), as each context is equally "close" to the generated tokens.

Figure 1: **Overview of our approach.** Context-augmented generation leverages additional contexts to improve LLM response quality to user queries. Sequential encoding prefills selected context chunks as a long sequence during inference, leading to high latency from on-the-fly re-encoding and low accuracy due to context window limitations. Parallel encoding offers an alternative method to pre-compute more and longer contexts within the same positional range but results in worse performance. To address these challenges, we propose Adaptive Parallel Encoding (**APE**) to re-align the attention weight distribution of parallel encoding with sequential encoding via three training-free steps: shared prefix, scaling factor, and adaptive temperature, leading to fast, long, and accurate CAG systems in real-world applications.

Despite these advantages, parallel encoding leads to significant performance degradation across multiple scenarios as shown in Figure 2, with average declines of 4.9% (despite using 2-10× more contexts) and 49.0%, respectively. While prior works (Sun et al., 2024; Yen et al., 2024) have attempted to correct this with fine-tuning, these methods continue to exhibit reduced accuracy in reasoning tasks. This performance drop can be attributed to the limited generalization ability from simple to complex tasks.

However, our results in Figure 2 also reveal that parallel encoding holds promise, as LLMs can still generate reasonable responses due to their inherent alignments with sequential encoding. Based on this observation, we aim to strengthen these alignments while addressing the remaining discrepancies to achieve more accurate parallel encoding. Our insight from Figure 3 and Figure 4 is that *KV states from independent contexts can be naturally merged into one sequence due to their similarity in direction and magnitude, attributed to the presence of an attention sink* (Xiao et al., 2023). This initial alignment influences the direction of following KV states, which consistently keep a similar large angle with the sink tokens (see Figure 5). Based on this observation, we can reduce our challenges to addressing residual misalignments, which manifest as anomalous distributions at the initial and recent positions within each context.

Motivated by these insights, we propose Adaptive Parallel Encoding (**APE**), which aligns the distribution of attention weights between sequential encoding and parallel encoding. APE enables accurate and fast CAG, as demonstrated in Figure 1 (bottom). The main contribution of this paper is as follows:

- We systematically analyze the distribution properties of attention weights in parallel encoding, focusing on the direction and magnitude of KV states across various samples and positions. Our analyses identify alignments and residual misalignments between parallel encoding and sequential encoding.
- We propose APE to recover the accuracy of parallel encoding with three alignment steps: (i) Prepend a shared prefix to avoid the duplication of abnormal initial token distributions. (ii) Adjust a lower attention temperature to sharpen the distribution, focusing on important tokens. (iii) Apply a scaling factor to offset the increase in the magnitude of the LogSumExp value of attention scores from the context.
- We empirically show that (i) APE keeps 98% and 93% of the sequential encoding performance in RAG and ICL tasks, respectively. (ii) APE outperforms parallel encoding in RAG and ICL, yielding improvements of 3.6% and 7.9%, respectively. (iii) APE effectively scales to handle hundreds of contexts, approaching or even surpassing sequential encoding in many-shot scenarios. (iv) APE accelerates long-context generation, achieving up to a 4.5× speedup through 28× reduction in prefilling time.

## 2 BACKGROUND AND RELATED WORK

### 2.1 CONTEXT-AUGMENTED GENERATION

This work explores CAG problems using LLMs, where user queries are enhanced with additional contexts from external databases. CAG involves two scenarios: RAG (Asai et al., 2024; Gupta et al., 2024; Gao et al., 2023), which focuses on directly retrieving relevant information, and ICL (Dong et al., 2022; Wei et al., 2022), which emphasizes further acquiring emergent capabilities from in-context examples.

### 2.2 PARALLEL ENCODING

Next, we present the formulation of using parallel encoding in LLMs for CAG. Let $\mathcal{S}$ represent the input sequence including $N$ contexts $C_1,...,C_N$ and one query $Q$. Formally, this can be denoted as:

$$\mathcal{S} = \{\underbrace{s_{C_1,1},...,s_{C_1,l_1}}_{\text{Context 1}}, \underbrace{s_{C_2,1},...,s_{C_2,l_2}}_{\text{Context 2}},..., \underbrace{s_{C_N,1},...,s_{C_N,l_N}}_{\text{Context N}}, \underbrace{s_{Q,1},...,s_{Q,l}}_{\text{Query}}\}. \tag{1}$$

For simplicity, we can express this as: $\mathcal{S} = \{S_{C_1}, S_{C_2},...,S_{C_N}, S_Q\}$. Given two models $\Theta_{\text{Enc}}$ and $\Theta_{\text{Dec}}$ (which may be identical), a response $\mathcal{R}$ is generated to input $\mathcal{S}$ using parallel encoding through two steps:

**Pre-caching Contexts.** The initial step involves encoding and caching the KV states for each context independently using $\Theta_{\text{Enc}}$. For a given context $S_{C_i}$, we compute the KV states as $(K_{C_i}, V_{C_i}) = \Theta_{\text{Enc}}(S_{C_i})$ and store them for subsequent direct use, where $K_{C_i} = \{k_{C_i,1},...,k_{C_i,l_i}\}$ and $V_{C_i} = \{v_{C_i,1},...,v_{C_i,l_i}\}$.

**Generating Response.** Next, we input the query and load all relevant KV states to generate response: $\mathcal{R} = \Theta_{\text{Dec}}(S_Q, K_C, V_C)$, where $K_C$, $V_C$ are subsets of $\{K_{C_1},...,K_{C_N}\}$ and $\{V_{C_1},...,V_{C_N}\}$, respectively.

Parallel encoding significantly improves efficiency compared to sequential encoding by reducing complexity from $O((l_1+...+l_N+l_Q)^2)$ (i.e., quadratic) to linear with respect to context length. With pre-caching, the cost becomes $O((l_1+...+l_N+l_Q)\cdot l_Q)$. In the absence of pre-caching, the complexity is $O(\max(l_1^2,...,l_N^2) + ((l_1+...+l_N+l_Q)\cdot l_Q))$, which is efficient for multiple contexts of similar length.

Prior parallel encoding approaches vary in their design of $\Theta_{\text{Enc}}$ and $\Theta_{\text{Dnc}}$. Parallel Context Windows (PCW) (Ratner et al., 2022) directly employs pre-trained LLMs as both, resulting in significant performance drops. Block-Attention (Sun et al., 2024) further fine-tunes the model, successfully recovering performance in RAG tasks. Alternatively, CEPE (Yen et al., 2024) and FocusLLM (Li et al., 2024) train new Transformer-based encoders using encoder-only and decoder-only architectures, respectively. These methods also differ in $\Theta_{\text{Dec}}$: CEPE trains additional cross-attention layers for processing contexts, whereas other methods directly input the context into original self-attention layers. While these trainable methods show promising results in RAG tasks, challenges remain regarding their training overheads and generalization abilities to more complex ICL scenarios. Moreover, applying parallel encoding in CAG can be viewed as a kind of memory-augmented neural networks (Burtsev et al., 2020; De Jong et al., 2021; Févry et al., 2020), where external memory is directly stored into KV states.

### 2.3 ATTENTION MECHINISM

In a standard $\text{Softmax}$ Attention, we attend the query to all KV states using the following formula:

$$O = \text{Softmax}(\frac{QK^T}{\sqrt{d}})V \quad Q \in \mathbb{R}^{n \times d} \quad K,V \in \mathbb{R}^{m \times d}, \tag{2}$$

where $Q$ is the query state, and $K$ and $V$ denote the key and value states, respectively. Previous research has revealed several significant insights into the distribution of attention weights (i.e., $\text{Softmax}(\frac{QK^T}{\sqrt{d}})$). These findings serve as the foundation for parallel encoding in LLMs, motivating our design of APE.

**Attention Sink.** StreamingLLM (Xiao et al., 2023) identifies the presence of an "attention sink" in LLMs – a token that receives a significantly higher attention score than other tokens but provides limited semantic information. It observes that the attention sink only exists in the initial token.

**Position Embedding.** To effective process sequential input, LLMs necessitate the explicit encoding of positional information. Common techniques include absolute positional embedding Vaswani (2017); Devlin (2018) and relative positional encoding (Su et al., 2024; Press et al., 2021). However, the introduction of position embedding not only limits the context window to the training length (Chen et al., 2023), but also results in the "lost in the middle" phenomenon (Liu et al., 2024a), where LLMs struggle to produce correct answers when relevant information is located in the middle of the context.

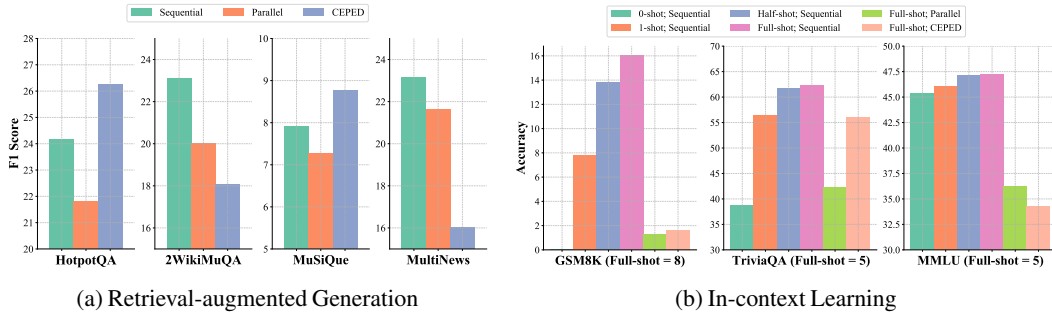

(a) Retrieval-augmented Generation (b) In-context Learning

Figure 2: Comparison of sequential encoding, parallel encoding, and CEPED in RAG and ICL scenarios. Parallel encoding and CEPED dramatically degrade performance, especially for math reasoning.

In this section, we evaluate sequential encoding, parallel encoding, and CEPE-Distilled (CEPED) (Yen et al., 2024) using the LLaMA-2-7B-CHAT model.[1] on various RAG and ICL tasks in Figure 2. Our analysis yields several key observations of parallel encoding. In Section 3.1, we explain the limitation of trainable approaches in their generalization abilities to complex tasks. Then, we explore why parallel encoding does not completely break down, despite LLMs being trained sequentially in Section 3.2. This analysis illuminates both alignments and misalignments between parallel and sequential encoding.

### 3.1 TRAINABLE APPROACHES ARE ONLY EFFECTIVE FOR EASY TASKS.

In Figure 2, we compare the performance of different context encoding methods on RAG and ICL tasks, with detailed setups described in Appendix B. Our analysis of the long-context RAG ability on LongBench (Bai et al., 2023) is showcased in Figure 2a. Despite accessing larger context windows, CEPED only surpasses the sequential baseline in 2/3 QA tasks, and it even notably underperforms parallel encoding in the Summarization task (MultiNews), which requires synthesizing information from the entire context. We hypothesize that CEPED cannot process complex tasks since the encoder and decoder are only trained unlabeled pre-training corpus, without instruction-tuning on high-quality QA samples. This is further supported by the results of ICL tasks (see Figure 2b), where CEPED performs on par with 1-shot sequential encoding baseline on TriviaQA but falls short of it on GSM8K and MMLU, despite using much more examples, respectively. The latter involves reasoning steps hard for the ill-trained model to understand. In conclusion, fine-tuning models to improve parallel encoding requires (i) more diverse and labeled data, and (ii) resource-intensive instruction-tuning (e.g., SFT or RLHF (Ouyang et al., 2022)). Given this unfavorable trade-off between training costs and model capabilities, we propose developing a training-free solution for better parallel encoding performance.

### 3.2 COMPARING PARALLEL ENCODING AND SEQUANTIAL ENCODING.

In Figure 2, we surprisingly observe that parallel encoding, despite reducing performance, can generate reasonable responses without further modifications. This is non-trivial as contexts are encoded into KV states separately, with no guarantee that these states can be compared or combined. However, our analysis reveals that the attention mechanism in LLMs naturally builds an inherent alignment across independent contexts similar to sequential encoding, despite the change in context and position. To clarify this, Figure 3 focuses on the impact of the attention sink (Xiao et al., 2023), where we visualize the direction (cosine similarity) of KV states for different samples and positions. In Figure 4, we further visualize the distribution of different components in the Softmax attention, resulting in several findings.

**Key states from different contexts are similar.** In Figure 3a and 3b, we measure the cosine similarity of the key states for different initial tokens for the LLaMA-3-8B-INSTRUCT and MISTRAL-7B-INSTRUCT-V0.3 models, which consistently yields a value close to 1. This indicates that the direction of the initial key state remains largely invariant across different inputs. Figure 3c and 3d further analyze the similarity be-

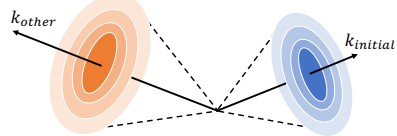

Figure 5: Geometry of Key States.

tween the initial key states and their subsequent states, where we observe comparable negative values from different positions. Therefore, the angles between the initial key states and their subsequent states are similar and significantly larger than the angles between different initial key states, as visualized in Figure 5. This suggests that the direction of key states remains relatively consistent across contexts, as they are primarily decided by the initial key states, which exhibit similar directions across examples. These findings, combined with the small variance in magnitude across examples in Figure 4b, show that key states from different contexts share similar directions and magnitudes, making them comparable.

---

[1]We use LLaMA-2 for CEPED, as it is the only supported model. For other analyses, we employ LLaMA-3.

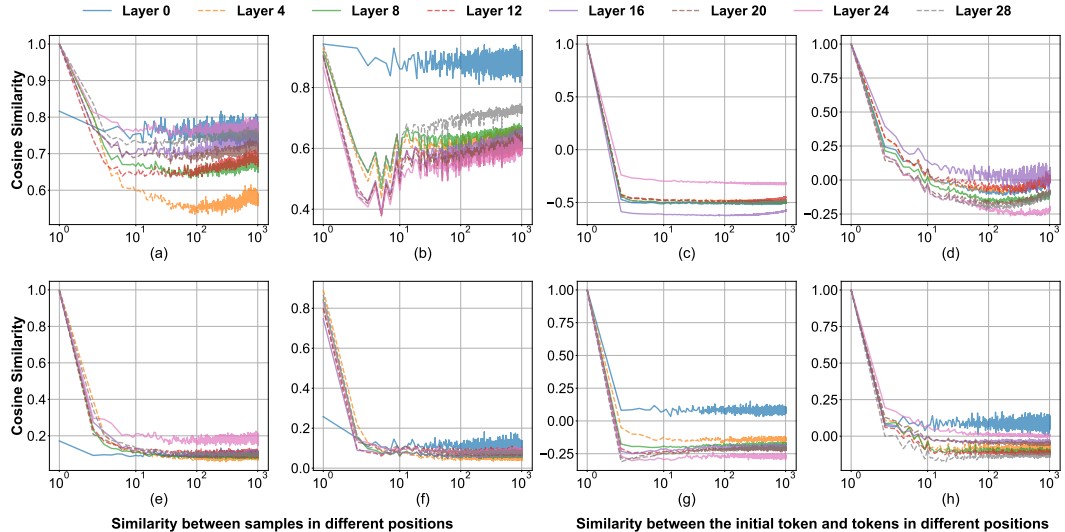

Figure 3: **Top Left:** Both LLAMA-3-8B-INSTRUCT (a) and MISTRAL-7B-INSTRUCT-V0.3 (b) states exhibit a cosine similarity larger than 0.9 for the key states from distinct initial tokens. **Top Right:** Initial token's key states show similar negative values to those from other positions for LLAMA-3-8B-INSTRUCT (c) and MISTRAL-7B-INSTRUCT-V0.3 (d) models. **Bottom:** Value states exhibit patterns similar to those observed in key states. The X-axis shows positions of key states on a logarithmic scale.

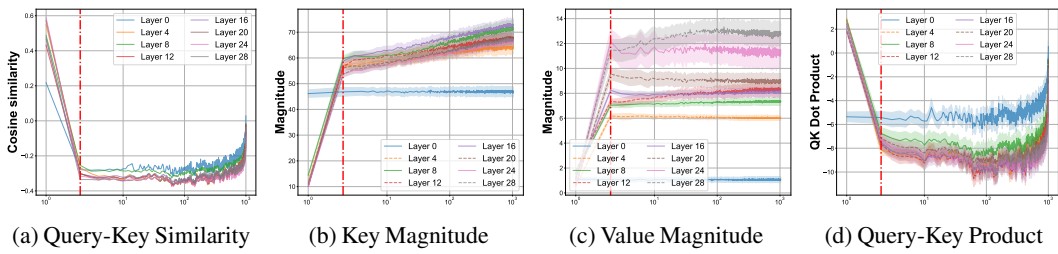

(a) Query-Key Similarity     (b) Key Magnitude     (c) Value Magnitude     (d) Query-Key Product

Figure 4: **(a)** The cosine similarity between query and key states increases as the distance between their positions decreases. **(b)** The magnitudes of key states show a slowly upward trend as position increases. **(c)** The magnitude of value states remain constant across positions. **(d)** Query-key dot products keep consistently low values except at initial and recent positions. A red dashed line marks the anomalous region for the first two tokens in all subfigures. The X-axis shows positions of key and value states on a log scale. Results are measured on the Hotpot QA dataset using the LLAMA-3-8B-INSTRUCT model.

To further understand this key observation, we conducted a small experiment using the LLAMA-3-8B-INSTRUCT model on the HotPotQA dataset. Our analysis involved applying rotations of varying degrees around random axes to the key states of initial tokens. For parallel encoding, we explored two rotation modes: one using the same rotation axis across different contexts, and another employing random rotation axis for each context. Figure 6 presents our findings, revealing that sequential encoding maintains performance across various rotation degrees. In contrast, both modes in parallel encoding deteriorate when

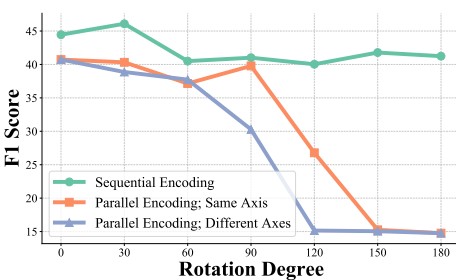

Figure 6: Rotation Analysis

rotations exceed 150 degrees. This effect arises from the duplication of initial tokens' key states, which intensifies the impact of our rotations. Notably, using separate axes for each context leads to an earlier breakdown beginning at 90 degrees. This disrupts the directional similarity of key states with different initial tokens (i.e., $k_{\text{initial}}$) in Figure 5 and enlarges the angle between key states from different contexts.

**Values states from different contexts can be combined.** In equation 2, value states are combined through a weighted summation in the attention calculation, where the Softmax operator would normalize the weights (i.e., attention scores) of all value states to sum to 1. This normalization indicates that the magnitude of current value states is determined solely by those from previous positions,

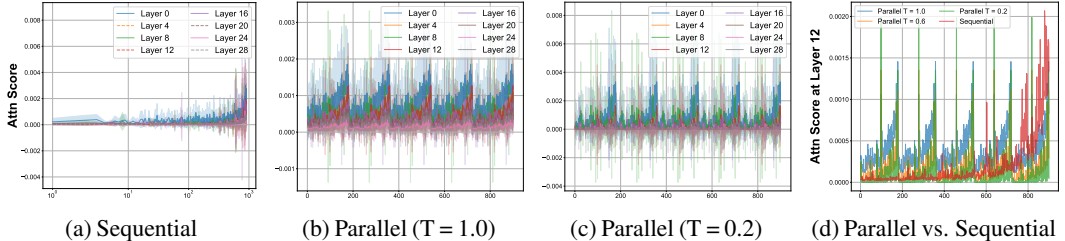

|  (a) Sequential | (b) Parallel (T = 1.0) | (c) Parallel (T = 0.2) | (d) Parallel vs. Sequential |

Figure 7: **(a)** Sequential encoding allocates high attention scores to neighboring tokens. **(b)** Parallel encoding distributes attention scores more uniform across neighboring tokens from all contexts. **(c)** Adjusting the temperature $T$ sparsifies the distribution. **(d)** After adjustment, the distribution in parallel encoding becomes similar to sequential encoding. The X-axis represents token positions.

resulting in a similar $L^2$ norm across positions, as shown in Figure 4c. Additionally, the small variance demonstrates that the magnitudes are comparable among samples. This, coupled with a similar direction across samples and positions in Figure 3 (Bottom), indicates that the value states can be combined.

**Opportunities for improvement.** Previous analyzes demonstrate that key and value states exhibit a natural alignment across contexts both in direction and magnitude for most positions. However, residual misalignments in Figure 4 still severely reduce performance. We summarize them as follows:

- In Figure 4, we observe a notable discrepancy in direction and magnitude for the first few positions, leading to large QK dot products in Figure 4d. This is designated as an abnormal region in the context.
- Figure 4d shows dot products between the final query state and all key states, revealing a notable increase when states are close to each other in position due to the larger cosine similarity in Figure 4a.

## 4 ADAPTIVE PARALLEL ENCODING

With all the lessons learned in Section 3, we will design our Adaptive Parallel Encoding (**APE**) to address the observed misalignments. APE enables a seamless shift to parallel encoding without requiring retraining, while maintaining the majority of the model's capabilities. Our approach adaptively aligns the distribution of attention weights between sequential encoding and parallel encoding through a three-step procedure as showcased in Figure 1 (Bottom), thereby boosting both efficiency and accuracy.

### 4.1 PREPENDING SHARED PREFIX.

Figure 4 illustrates that the distribution of various components for the first few tokens differ significantly from those of subsequent tokens. This discrepancy poses a challenge when encoding contexts in parallel from the beginning, as it would result in the duplication of these abnormal KV states. To address this issue, we propose a simple yet effective solution: prepending a shared prefix to all contexts. This approach ensures that these KV states appear only once in each generation step. In practice, the choice of prefix varies with the model and task. We use existing system prompts and instructions as the shared prefix when available. Otherwise, we will insert a few newline characters (i.e., "\n") before all contexts.

### 4.2 ADJUSTING ATTENTION TEMPERATURE.

In Figure 4d, the value of QK dot products increases as the relative distance decreases, with a notably sharper rise when the distance approaches zero. To show its impact on parallel encoding, we set a 50-token prefix and query, encoding the remaining 900 tokens sequentially or in five parallel chunks, with attention distributions shown in Figure 7. Comparing Figure 7b with 7a, duplicating neighboring KV states in parallel encoding will disperse the query's attention to multiple contexts, resulting in a more uniform attention distribution. We adjust the attention temperature $T$ to a value less than 1 to refocus on the most relevant tokens. The comparison between different $T$ is shown in Figure 7c and 7d.

### 4.3 ADDING SCALING FACTOR.

While adjusting the temperature sharpens the attention distribution among context tokens but also alters the overall attention allocated to them, as shown by the LogSumExp value with different $T$ in Figure 8. Specifically, when the sum of the original QK dot product values is significantly greater than 0, reducing $T$ amplifies these positive values, resulting in an increased, positive LogSumExp value. Conversely, when the sum is closer to 0, lowering $T$ has a stronger effect on the negative QK dot products, leading to a decreased, negative LogSumExp value. These effects generally increase the absolute value of LogSumExp. To mitigate this, we add a scaling factor $S < 1$ to directly reduce the absolute value.

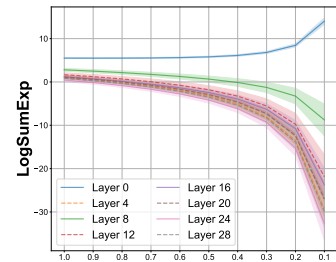

Figure 8: **Parallel w/ Diverse $T$.**

## 4.4 FORMULATION.

Given these three steps, we can formulate the modified attention in APE. We begin with the standard $\mathrm{Softmax}$ attention, where $Q$, $K$, and $V$ are the query, key, and value states from the input or output:

$$O = \mathrm{Softmax}\left(\frac{Q[K_{C_1}^\top,...,K_{C_N}^\top,K^\top]}{\sqrt{d}}\right) \times [V_{C_1},...,V_{C_N},V] \tag{3}$$

$$= \frac{[A_{C_1},...,A_{C_N},A]}{\sum_{i=1}^{N}\sum_{j=1}^{l_{C_i}} a_{C_i,j} + \sum_{j=1}^{l} a_j} \times [V_{C_1},...,V_{C_N},V], \tag{4}$$

where $A_{C_i} = [\exp\frac{Qk_{C_i,1}^\top}{\sqrt{d}},...,\exp\frac{Qk_{C_i,l_{C_i}}^\top}{\sqrt{d}}]$ and $a_{C_i,j} = \exp\frac{Qk_{C_i,j}^\top}{\sqrt{d}}$. Similar for $A$ and $a_j$.

After incorporating the proposed changes, the formula for our refined attention calculation becomes:

$$O' = \frac{[A_P, A'_{C_1},...,A'_{C_N}, A]}{\sum_{j=1}^{l_P} a_{P,j} + (\sum_{i=1}^{N}\sum_{j=1}^{l_{C_i}} a'_{C_i,j})^S + \sum_{j=1}^{l} a_j} \times [V_P, V_{C_1},...,V_{C_N}, V], \tag{5}$$

where $A'_{C_i} = [\exp\frac{Qk_{C_i,1}^\top}{T\sqrt{d}},...,\exp\frac{Qk_{C_i,l_{C_i}}^\top}{T\sqrt{d}}] \cdot (\sum_{i=1}^{N}\sum_{j=1}^{l_{C_i}} a'_{C_i,j})^{S-1}$ and $a'_{C_i,j} = \exp\frac{Qk_{C_i,j}^\top}{T\sqrt{d}}$.

$A_P$ represents the attention weights for the shared prefix while $A$ denotes that for query and generated tokens. The attention temperature $T$ and the scaling factor $S$ for the context are less than 1. Appendix D provides a detailed deduction of this formula. All these modifications are compatible with fast attention implementations such as flash attention (Dao et al., 2022) by computing the context and non-context KV states separately and merging them as the attention output, which only incur a negligible overhead.

We tune hyperparameters on a validation set with greedy search. If no prefix is provided, we begin by adding two "\n" and increase the prefix length by 10, 20, and 40. $S$ and $T$ are searched in the ranges [0.1, 1.0] using 0.1 step sizes. We use $S \cdot T$ instead of $S$ as the scaling factor to simplify our search.

## 5 EXPERIMENTS

In this section, we present the effectiveness and efficiency of APE, focusing on CAG scenarios such as RAG and ICL. Therefore, we do not include comparisons with long-context LLMs here. Specifically,

- In Section 5.1, APE can maintain 98% of the accuracy on ChatRAG-Bench compared to sequential encoding when retrieving the same text as input. Furthermore, it improves performance by 3.3% on average for RAG on LongBench through its ability to retrieve and process more and longer contexts.
- In Section 5.2, APE outperforms parallel encoding by 7.9% on average in ICL tasks. Moreover, APE keeps 93% of accuracy achieved by sequential encoding when using the same number of examples.
- In Section 5.3, APE scales to many-shot CAG tasks, effectively encoding hundreds of texts in parallel.
- In Section 5.4, APE can achieve up to $4.5\times$ faster inference through $28\times$ reduction in prefilling time.

## 5.1 RETRIEVAL-AUGMENTED GENERATION.

For RAG tasks, we validate that APE retains most of the sequential encoding capability while accommodating more and longer contexts, mitigating retrieval errors and outperforming encoding baselines.

### 5.1.1 RETRIEVAL FOR MULTI-TURN QUESTION ANSWERING.

**Setup.** APE is evaluated on five conversational QA tasks using ChatRAGBench (Liu et al., 2024b). For each query, 1–100 text chunks are prepared, with retrievers employed to select the top-$n$ most relevant chunks as input. We utilize three different retrievers of varying quality: Contriever (Izacard et al., 2021), GTE-base Li et al. (2023), and Dragon-multiturn Liu et al. (2024b), and retrieve up to the top-5 chunks for evaluation. The base model used is LLAMA3-CHATQA-1.5-8B. To measure performance drop after our alignment steps, the same retrieved texts are used for both APE and sequential encoding.

**Results.** Table 1 shows that switching from sequential encoding to APE results in performance drops of 0.51%, 0.92%, and 1.14% across different retrievers, respectively. Notably, while this drop increases with retriever quality, APE still keeps 97% of the sequential encoding performance for the best retriever. Moreover, APE can directly input all texts without retrieval process, achieving superior performance.

### 5.1.2 RETRIEVAL FOR LONG-CONTEXT UNDERSTANDING.

**Setup.** Our evaluation involves eight tasks on LongBench (Bai et al., 2023). Given the long context, we split it into chunks with a default size of $M$ words, employ Contriever (Izacard et al., 2021) to compute the embeddings of the text chunks and the query, and retrieve the top-$N$ chunks according to the cosine similarity of their embeddings to the query embedding. $M$ and $N$ vary across different

Table 1: Comparison between APE and sequential encoding using three retrievers on ChatRAG-Bench.

| Method | INSCIT | Doc2Dial | TopicCQA | Qrecc | QuAC | Average |
|---|---|---|---|---|---|---|
| Contriever, Sequential | 19.97 | 23.85 | 30.49 | 46.75 | 26.57 | 29.53 |
| Contriever, APE | 19.88 | 23.28 | 28.84 | 46.28 | 26.80 | 29.02 |
| Δ | -0.09 | -0.57 | -1.65 | -0.47 | +0.23 | -0.51 |
| GTE-base, Sequential | 21.58 | 32.35 | 33.41 | 46.54 | 30.69 | 32.91 |
| GTE-base, APE | 20.85 | 30.99 | 31.92 | 45.83 | 30.35 | 31.99 |
| Δ | -0.73 | -1.36 | -1.49 | -0.71 | -0.34 | -0.92 |
| Dragon-multiturn, Sequential | 25.42 | 36.27 | 36.10 | 49.01 | 35.12 | 36.38 |
| Dragon-multiturn, APE | 23.84 | 34.93 | 33.80 | 48.70 | 34.92 | 35.24 |
| Δ | -1.58 | -1.34 | -2.30 | -0.31 | -0.20 | -1.14 |
| All texts, APE | **27.22** | 36.13 | 35.72 | **49.15** | **35.70** | **36.78** |

Table 2: Comparison between APE and baselines on LongBench across different models using RAG. C denotes Contriever, and $M \times N$ indicates retrieval of the top-$N$ chunks, each containing $M$ words.

| Model | MuSiQue | Qasper | 2WikiMQA | DuRead | HotpotQA | NarratQA | MFQA_zh | MFQA_en | Avg. |
|---|---|---|---|---|---|---|---|---|---|
| LLAMA-3-8B-INSTRUCT | 20.70 | 41.05 | 30.02 | 9.55 | 45.90 | 20.98 | **58.54** | 45.04 | 33.97 |
| C200×20, Sequential | **27.93** | **42.71** | 38.35 | 12.65 | 49.60 | 22.78 | 57.82 | **48.94** | 37.60 |
| C4000×20, PCW | 18.82 | 42.59 | 40.99 | 21.57 | 47.09 | 23.29 | 54.40 | 45.05 | 36.73 |
| C4000×20, APE | 26.19 | 42.32 | **44.43** | **23.13** | **49.71** | **30.71** | 55.03 | 45.41 | **39.62** |
| MISTRAL-7B-INSTRUCT-V0.3 | 10.05 | 31.08 | 22.12 | 17.68 | 32.09 | 19.68 | 32.03 | 40.38 | 25.64 |
| C200×20, Sequential | 11.58 | 21.98 | 24.44 | 20.80 | 32.79 | 16.06 | 34.43 | 38.40 | 25.06 |
| C4000×20, PCW | 17.58 | 35.57 | 32.97 | 18.70 | 37.05 | 14.10 | 34.69 | 40.14 | 28.85 |
| C4000×20, APE | **20.30** | **36.81** | **34.37** | **21.89** | **42.33** | **20.49** | 40.20 | **44.03** | **32.55** |
| GEMMA-2-9B-IT | 22.57 | 39.99 | 48.06 | 27.40 | 47.49 | 23.11 | 50.81 | 45.35 | 38.10 |
| C200×10, Sequential | 30.69 | 42.86 | **53.55** | 28.04 | 52.05 | 24.45 | 50.25 | 48.34 | 41.28 |
| C2000×20, PCW | 26.27 | 46.69 | 47.59 | 23.43 | 48.95 | 27.11 | **56.69** | 49.81 | 40.82 |
| C2000×20, APE | **33.38** | **47.72** | 49.49 | **28.43** | **56.62** | **30.41** | 56.52 | **50.84** | **44.18** |
| LLAMA-3.1-8B-INSTRUCT | 22.18 | **46.81** | 40.58 | **34.61** | 43.97 | 23.08 | 61.60 | 51.89 | 38.98 |
| 128K, Sequential | 28.35 | 47.20 | 40.81 | 33.34 | 53.46 | 30.57 | 61.97 | 53.25 | 42.24 |
| C200×20, Sequential | **30.62** | 42.33 | 44.39 | 33.51 | 49.97 | 23.87 | 56.87 | **55.14** | 40.22 |
| C4000×20, PCW | 21.23 | 41.52 | 44.87 | 31.11 | 49.47 | 19.98 | 60.90 | 51.19 | 38.44 |
| C4000×20, APE | 26.88 | 43.03 | **50.11** | 32.10 | **55.41** | **30.50** | **62.02** | 52.51 | **42.86** |

encoding methods. We compare with sequential encoding with and without RAG, and PCW, using LLAMA-3-8B-INSTRUCT (Dubey et al., 2024), MISTRAL-7B-INSTRUCT-V0.3 (Jiang et al., 2023), GEMMA-2-9B-IT (Team et al., 2024), and LLAMA-3.1-8B-INSTRUCT as base models.

**Results.** In Table 2, APE consistently improves performance across all models, achieving a 5.6% average gain over base models and outperforming sequential RAG baselines by 3.3% through retrieval of more and longer contexts. The superior performance over PCW further showcases the effectiveness of our alignments. Notably, APE surpasses the 128K-context variant of LLAMA-3.1-8B-INSTRUCT by placing retrieved texts within the 8K context window, mitigating the "lost in the middle" phenomenon.

### 5.2 IN-CONTEXT LEARNING

**Setup.** We evaluate APE on three ICL tasks using the LM Evaluation Harness (Gao et al., 2024):: GSM8K (8-shot) (Cobbe et al., 2021a), TriviaQA (5-shot) (Joshi et al., 2017), and MMLU (5-shot) (Hendrycks et al., 2020a).Experiments use the same base models as in our LongBench evaluations.

**Baselines.** We compare parallel encoding (PCW) to show the improvement from APE's alignment steps. Sequential encoding with varying numbers of shots (i.e., 1-shot, half-shots, and full-shots) are also employed to measure the gap from the ideal scenarios. All methods can access all input examples.

**Results.** In Figure 9, APE significantly surpasses parallel encoding with average improvements of 15.4% on GSM8K, 4.7% on TriviaQA, and 3.5% on MMLU. When compared with the 1-shot sequential baseline with comparable context length, our method consistently yields superior results. Moreover, APE achieves better performance than half-shot sequential encoding in 8/12 settings and preserve 93% accuracy comparing to the full-shot sequential encoding. Additionally, our results suggest that the LLAMA family exhibits enhanced compatibility with parallel encoding, potentially due to the stronger directional alignment of initial tokens from different contexts (see Figure 3a). Across different tasks, the performance gap between APE and full-shot sequential encoding is the largest on GSM8K, suggesting that while APE maintains most capabilities, its effectiveness may decrease as task complexity increases.

### 5.3 MANY-SHOT CONTEXT-AUGMENTED GENERATION

**Setup.** To demonstrate the scalability of APE, we employ four RAG and ICL tasks from the LOFT benchmark (Lee et al., 2024), each involving hundreds of texts that provide additional information. We

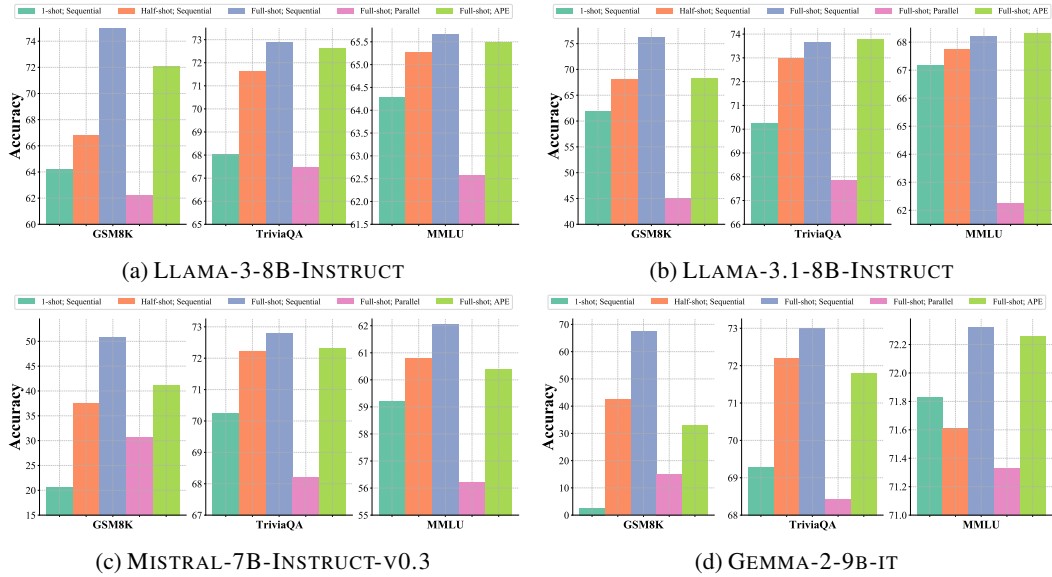

(a) LLAMA-3-8B-INSTRUCT

(b) LLAMA-3.1-8B-INSTRUCT

(c) MISTRAL-7B-INSTRUCT-v0.3

(d) GEMMA-2-9B-IT

Figure 9: Performance comparison of APE, parallel encoding, and sequential encoding on ICL tasks.

Table 3: Comparison between APE and sequential encoding in various many-shot RAG and ICL tasks.

| Method | Retrieval-augmented Generation | | | | In-context Learning | | | |
|---|---|---|---|---|---|---|---|---|
| | ArguAna | FEVER | NQ | SciFact | Date | Salient | Tracking7 | Web |
| Sequential, Zero-shot | 11.15 | 7.78 | 17.78 | 7.74 | 20.00 | 8.89 | 1.12 | 8.89 |
| Sequential, Few-shot | 11.20 | 9.78 | 17.81 | 9.49 | 36.64 | 38.89 | 6.67 | 38.89 |
| Sequential, Half-shot | 15.34 | 13.12 | 19.64 | 16.12 | 45.55 | 42.22 | 8.89 | 55.56 |
| Sequential, Full-shot | 12.84 | 14.19 | **24.54** | **16.88** | **46.67** | **46.67** | **8.89** | **58.89** |
| APE, Full-shot | **16.32** | **14.70** | 21.91 | 15.72 | 43.33 | 45.55 | **8.89** | **58.89** |

employ LLAMA-3.1-8B-INSTRUCT as our base model to compare APE with sequential encoding, both applied to the same many-shot long-context inputs. The total context lengths for RAG and ICL tasks are 128K and 32K, respectively. We also include the zero-shot, few-shot ($\leq 5$), and half-shot variants of sequential encoding as baselines. For metrics, F1 score and EM are used in RAG and ICL.

**Results.** As shown in Table 3, APE achieves performance comparable to sequential encoding when processing the same many-shot long-context inputs, showcasing its ability to efficiently encode hundreds of texts in parallel. Notably, for RAG tasks, it outperforms sequential encoding on ArguAna and FEVER. While APE is expected to reduce performance, it recovers this drop by positioning all texts close to the query. This mechanism addresses the "lost in the middle" problem in long-context LLMs. Additionally, for ICL tasks, APE can learn from examples as effective as sequential encoding.

### 5.4 EFFICIENCY EVALUATION

**Setup.** We measure the latency for sequential encoding, MInference (Jiang et al., 2024a), and APE on the Llama-3.1-8B-Instruct (Dubey et al., 2024) model. Our evaluation is conducted on an H100 GPU with batch sizes of 1 and 4. The query and generation lengths are fixed at 256 tokens, while the input context lengths ranged from 2K to 128K tokens. We employ VLLM (Kwon et al., 2023) as our inference engine and measure both prefilling time (Time to first token) and total inference time.

**Results.** Comparing to sequential encoding and MInference, APE can accelerate inference up to $4.5\times$ and $2.2\times$ respectively for long-context scenarios in Figure 10. Notably, for 128K-token contexts, APE reduces prefilling time by $28\times$ compared to MInference. APE's prefilling overhead exhibits linear scaling and consumes less than 10% of inference time, while baselines require over 50% as

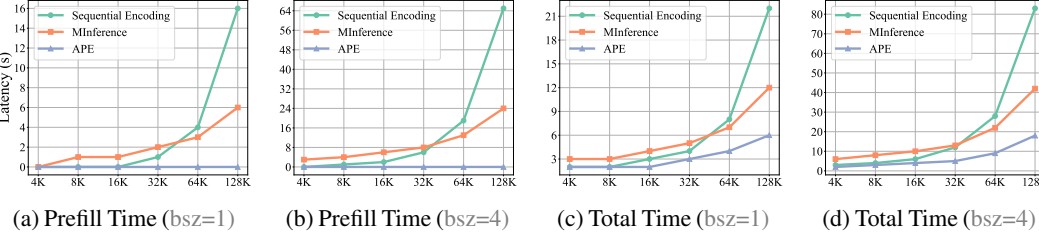

(a) Prefill Time (bsz=1)    (b) Prefill Time (bsz=4)    (c) Total Time (bsz=1)    (d) Total Time (bsz=4)

Figure 10: Latency on H100 GPU: [prefill and total time (s)]. The gray text in brackets is batch size.

Table 5: Performance and latency comparison using the LLAMA-3-8B-INSTRUCT model on CRAG.

| Task | Model | Latency (ms) | Accuracy (%) | Hallucination | Missing | Score$_a$ |
|---|---|---|---|---|---|---|
| LLM only | LLAMA-3-8B-INSTRUCT | 682 | 22.14 | 48.97 | 28.90 | -26.83 |
| Task 1 | LLAMA-3-8B-INSTRUCT | 1140 | 23.28 | 29.49 | 47.22 | -6.21 |
| | +APE | 1054 | **25.53** | **21.30** | **37.93** | **-0.41** |
| Task 2 | LLAMA-3-8B-INSTRUCT | 1830 | 24.46 | 28.38 | 47.15 | -3.92 |
| | +APE | 1672 | **27.04** | **18.74** | **37.32** | **2.16** |

Table 6: Performance comparison across different long-context tasks on LongBench (Bai et al., 2023).

| Method | NarratQA | Qasper | MultiFQA | GovReport | QMSum | LCC |
|---|---|---|---|---|---|---|
| LLAMA-3-8B-INSTRUCT | 19.32 | 32.83 | 43.38 | 27.89 | 22.40 | 53.22 |
| +APE | **26.87** | **39.14** | **59.12** | **29.10** | **23.08** | **66.09** |

| Method | RepoBench-P | HotpotQA | 2WikiMQA | MuSiQue | MultiNews | Average |
|---|---|---|---|---|---|---|
| LLAMA-3-8B-INSTRUCT | 38.15 | 44.24 | 21.01 | 20.47 | **23.63** | 31.50 |
| +APE | **49.43** | **50.11** | **28.06** | **25.79** | 22.40 | **38.11** |

context length increases. APE also demonstrates superior versatility across various settings, whereas MInference slows inference with additional overhead when processing short contexts and large batches.

# 6 ANALYSIS

In this section, we present analyses to answer the following research questions: **RQ1**: How does each component in APE contribute to the performance? **RQ2**: Can APE improve performance for real-world RAG tasks? **RQ3**: Can APE extend LLM context window size in long-context scenarios without RAG?

## 6.1 HOW DOES EACH COMPONENT IN APE CONTRIBUTE TO THE PERFORMANCE?

In Table 4, we conduct an ablation study to examine each alignment process in APE, including the shared prefix ($P$), attention temperature ($T$), and scaling factor ($S$). We present results averaged across the four models evaluated in Figure 9. Our findings indicate that incorporating each of these components can consistently improve performance across all tasks, with average improvements of 5.19%, 0.59%, and 2.07%, respectively. Among them, adjusting the attention temperature yields minimal performance gains without the complementary effect of the scaling factor.

Table 4: Ablation study of APE alignments on ICL tasks. $P$: shared prefix, $T$: attention temperature, $S$: scaling factor.

| $P$ | $T$ | $S$ | GSM8K | TriviaQA | MMLU |
|---|---|---|---|---|---|
| | | | 38.25% | 67.99% | 63.09% |
| ✓ | | | 50.42% | 70.76% | 63.70% |
| ✓ | ✓ | | 51.15% | 71.03% | 64.49% |
| ✓ | ✓ | ✓ | **53.62%** | **72.64%** | **66.62%** |

## 6.2 CAN APE IMPROVE PERFORMANCE FOR REAL-WORLD RAG APPLICATIONS?

In Table 5, we evaluate APE's performance in real-world RAG scenarios using the CRAG benchmark (Yang et al., 2024). Task 1 augments the model with several webpages, while Task 2 provides an additional knowledge graph as another retrieval source. In our experiments, the sequential encoding baseline is limited to retrieving 4K tokens of context, whereas APE can process 20 parallel segments of 4K tokens each. By incorporating significantly more external data during generation, APE consistently outperforms sequential encoding that has limited context sizes while reducing latency. Moreover, the improvement in Task 2 further shows the effectiveness of APE in merging text from multiple sources.

## 6.3 CAN APE EXTEND CONTEXT LENGTHS IN LONG-CONTEXT SCENARIOS WITHOUT RAG?

Table 6 examines the effectiveness of APE when processing a single long context input for the LLAMA-3-8B-INSTRUCT model on LongBench (Bai et al., 2023). To accommodate the long context within our APE, we split it into multiple segments with of less than 7,500 tokens, appending the final 500 tokens to the query. Our results indicate that APE enhances performance across 10/11 tasks, yielding an average improvement of 6.6% compared to the sequential encoding baseline with limited context window size.

# 7 CONCLUSION

This work explores parallel encoding in context-augmented generation, which can pre-cache KV states for fast inference and re-use positions for long context but lead to worse performance. To address this, we propose APE, a training-free method to enable accurate and fast CAG systems. It achieves this by aligning the attention distribution of parallel encoding with sequential encoding via three steps: shared prefix, adaptive temperature, and scaling factor. We show that APE improves accuracy and efficiency in various RAG and ICL scenarios while successfully scaling to process hundreds of chunks in parallel.

ACKNOWLEDGEMENT

This work is supported in part by NSF award CNS-2211882 and a gift from Qualcomm. We gratefully acknowledge additional support from Amazon, Intel, Li Auto, and Moffett AI. Additionally, we extend our gratitude to the authors of ChatQA (Liu et al., 2024b), Longbench (Bai et al., 2023), CRAG (Yang et al., 2024), LM Evaluation Harness (Gao et al., 2024), VLLM (Kwon et al., 2023), and MInference (Jiang et al., 2024a) for their valuable codebases, benchmarks, and models. We also sincerely thank Yixin Dong, Hanshi Sun, Zhuoming Chen for their insightful discussions and feedback.

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

## A    LIMITATIONS

While APE preserves the efficiency of parallel encoding and the effectiveness of sequential encoding through inference-time modifications to the attention distribution, its performance remains highly sensitive to hyperparameter selection, particularly the attention temperature $T$ and scaling factor $S$. In real-world applications, where contexts vary in length, quantity, and content, automatically aligning the distribution between sequential and parallel encoding during runtime presents a significant challenge.

## B    DETAILED EXPERIMENTAL SETUPS FOR SECTION 3.1

**RAG.** We evaluate four tasks from the LongBench dataset (Bai et al., 2023) that require processing multiple independent documents: HotpotQA (Yang et al., 2018), 2WikiMultihopQA (Ho et al., 2020), MuSiQue (Trivedi et al., 2022), and MultiNews (Fabbri et al., 2019). For the three QA tasks, we use the F1 score as the evaluation metric, while Rouge-L is used for the summarization task. Both parallel encoding and CEPED process each document independently using $\Theta_{\text{Enc}}$. If a document exceeds the window size of $\Theta_{\text{Enc}}$, these methods split it into multiple chunks before encoding. In contrast, sequential encoding handles lengthy inputs by truncating them from the middle to fit within the context window.

**ICL.** We select three few-shot learning tasks from LM Evaluation Harness (Gao et al., 2024) to evaluate the ICL ability of different encoding methods of different encoding methods: GSM8K (Cobbe et al., 2021b), TriviaQA Joshi et al. (2017), and MMLU (Hendrycks et al., 2020b). For parallel encoding and CEPED, each example is encoded independently, with all resulting KV states stored as input to $\Theta_{\text{Dec}}$. To thoroughly analyze these methods, we evaluate variants of sequential encoding with different numbers of shots—0-shot, 1-shot, half-shot, and full-shot, and compare with the full-shot parallel ones.

## C    MORE VISUALIZATIONS FOR SECTION 3.2

### C.1    SIMILARITY BETWEEN KEY STATES FROM DIFFERENT SAMPLES IN EACH POSITION.

In Figure 11, we show that key states across different layers maintain consistently high cosine similarity values for various initial tokens, with only the first layer exhibiting slightly reduced similarities. Our analysis reveals that LLAMA-3-8B-INSTRUCT and LLAMA-3.1-8B-INSTRUCT maintain nearly identical directions (cosine similarity approximately 1.0) for different tokens beyond the first layer, while MISTRAL-7B-INSTRUCT-v0.3 and GEMMA-2-9B-IT display substantial but comparatively lower similarities ranging from 0.8 to 0.9. These findings indicate inherent alignments across contexts while highlighting the potential for further improvements through the shared prefix in Section 4.1.

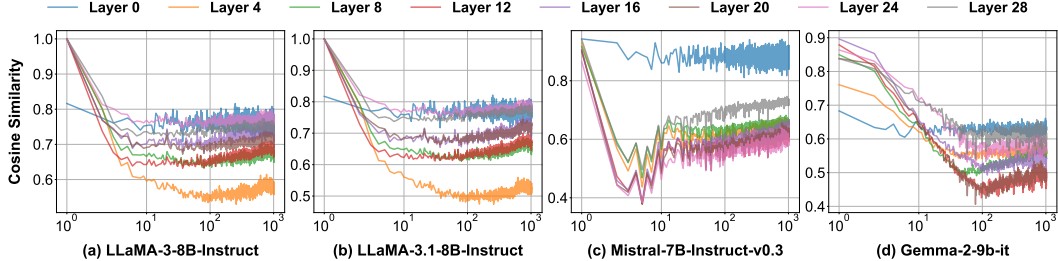

Figure 11: For all base models, key states from distinct initial tokens exhibit a cosine similarity larger than 0.8 for most layers, where the LLaMA family even approaches 1. In contrast, later tokens show markedly lower cosine similarity. The X-axis shows the positions of key states on a logarithmic scale.

### C.2    SIMILARITY BETWEEN VALUE STATES FROM DIFFERENT SAMPLES IN EACH POSITION.

Figure 12 illustrates that value states keep high cosine similarity across layers for various initial tokens, with two exceptions: the first layer in all models and some layers in GEMMA-2-9B-IT. This observation aligns with the requirement that GEMMA-2-9B-IT necessitates a system prompt for proper functioning.

### C.3    SIMILARITY BETWEEN THE INITIAL KEY STATES AND FOLLOWING KEY STATES.

Figure 13 illustrates how the cosine similarity between the initial and subsequent key states stabilizes as position increases. This similarity converges to a near-constant value for all models after 10 tokens.

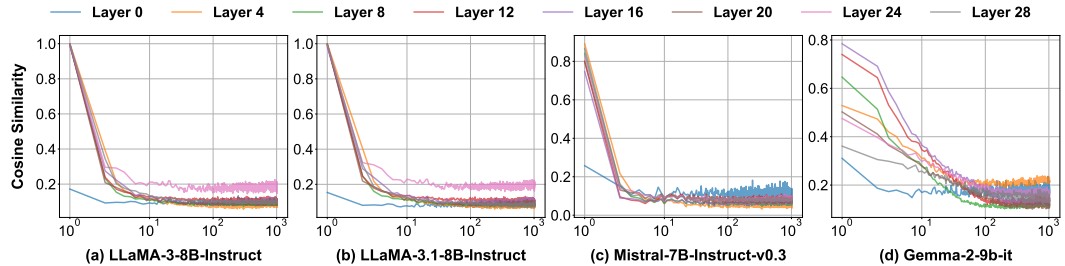

Figure 12: Across all four models, value states from distinct initial tokens exhibit higher cosine similarity than other positions, which is larger than 0.8, except for the first layer in all models and some layers in GEMMA-2-9B-IT. The X-axis displays the positions of value states on a logarithmic scale.

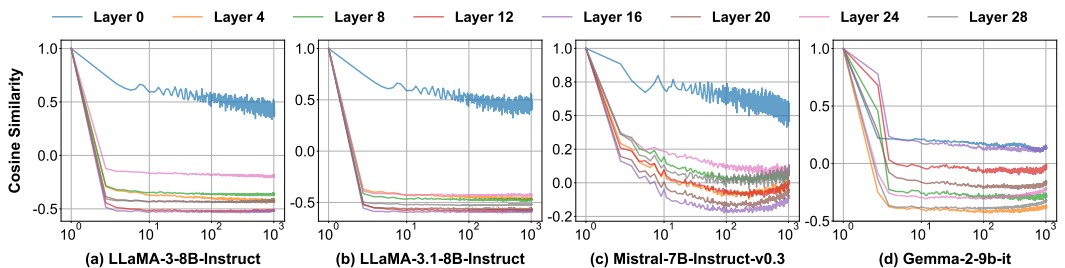

Figure 13: For all base models, the similarity between the initial key state and subsequent key states stabilizes as the position increases. The X-axis shows the positions of key states on a logarithmic scale.

## C.4 SIMILARITY BETWEEN THE INITIAL VALUE STATES AND FOLLOWING VALUE STATES.

Similar to key states, the value states exhibit a stable similarity between the initial token and subsequent tokens in Figure 14, with all models convergent to a near-constant value after approximately 10 tokens.

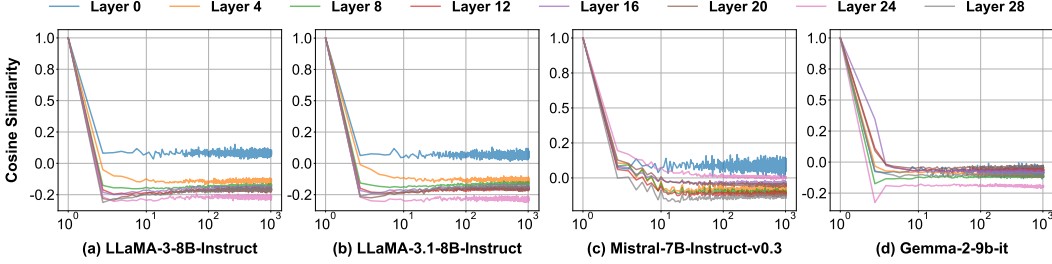

Figure 14: For all base models, the similarity between the initial value state and subsequent value states stabilizes as the position increases. The X-axis shows value states' positions on a logarithmic scale.

## C.5 SIMILARITY BETWEEN THE QUERY STATE AND PAST KEY STATES.

In Figure 15, the query states across all layers, and base models exhibit higher cosine similarity with the initial tokens. Additionally, neighboring positions tend to receive higher cosine similarity.

## C.6 MAGNITUDE OF KEY STATES FROM DIFFERENT POSITIONS.

Figure 16 illustrates that the magnitude of key states slowly increases with position, except for the first few tokens, which exhibit significantly smaller magnitudes.

## C.7 MAGNITUDE OF VALUE STATES FROM DIFFERENT POSITIONS.

In Figure 17, the value states across all positions exhibit a similar magnitude, except for the first few positions, which show a noticeable deviation. We indicate this region with a red dashed line.

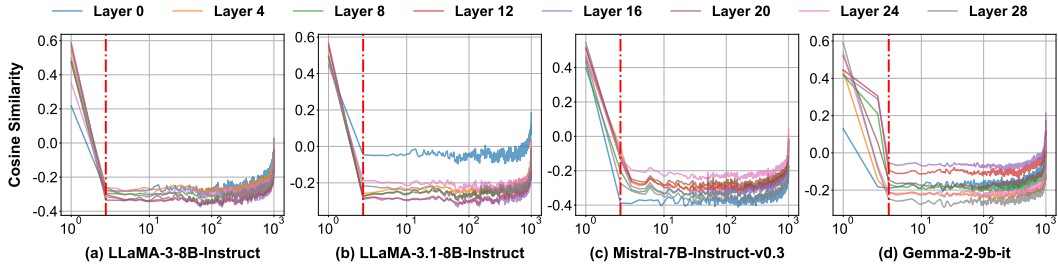

Figure 15: The cosine similarity between the query and key states stabilizes for most positions, except for the initial and recent positions. The X-axis shows the positions of key states on a logarithmic scale.

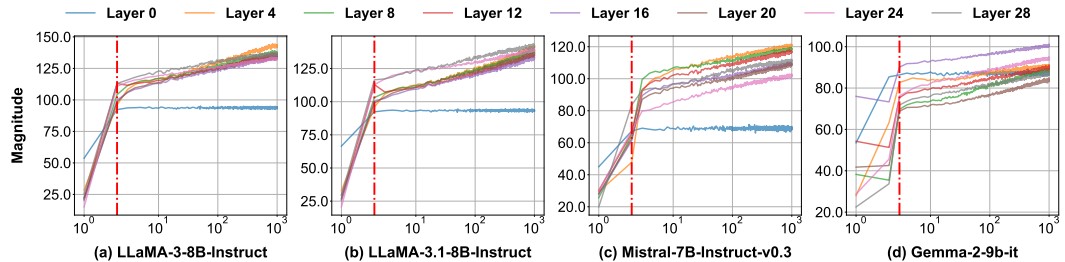

Figure 16: The magnitude of key states increases slowly with position, with a red dashed line marking the anomaly in initial tokens. The X-axis displays the positions of key states on a logarithmic scale.

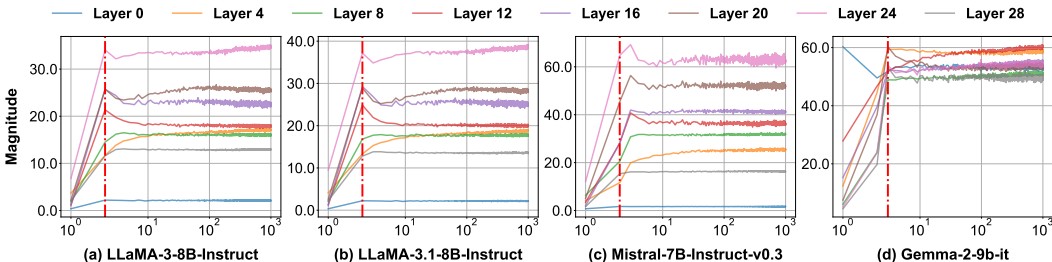

Figure 17: The magnitude of value states remains consistent, except for the first few positions highlighted by a red dashed line. The X-axis represents the positions of value states on a logarithmic scale.

### C.8 DOT PRODUCT BETWEEN THE QUERY STATE AND PAST KEY STATES.

In Figure 18, the query states across all layers, and base models exhibit larger dot product values with the initial tokens. Additionally, tokens at neighboring positions tend to receive higher values as well.

## D FORMAL DERIVATION OF APE

### D.1 HIERARCHICAL FORMULA FOR SOFTMAX ATTENTION.

We begin with the standard $\mathrm{Softmax}$ attention, where $Q$, $K$, and $V$ represent query, key, and value states. To distinguish between components from different sources, we use the subscript $C_i$ for elements from the context, while those without a subscript correspond to user queries or generated responses.

$$O = \mathrm{Softmax}\left(\frac{Q[K_{C_1}^\top, ..., K_{C_N}^\top, K^\top]}{\sqrt{d}}\right) \times [V_{C_1}, ..., V_{C_N}, V] \tag{6}$$

$$= \frac{[A_{C_1}, ..., A_{C_N}, A]}{\sum_{i=1}^{N}\sum_{j=1}^{l_{C_i}} a_{C_i, j} + \sum_{j=1}^{l} a_j} \times [V_{C_1}, ..., V_{C_N}, V], \tag{7}$$

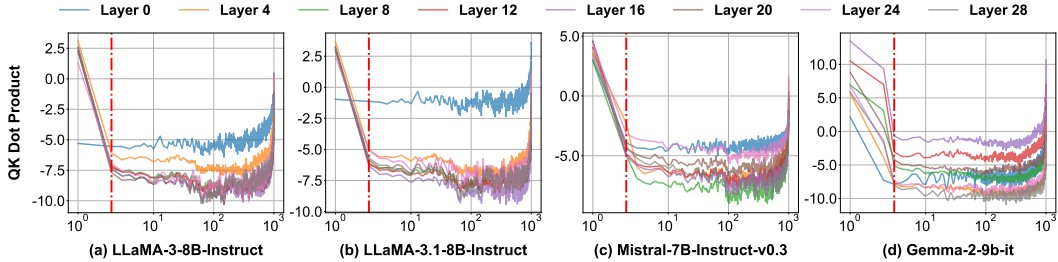

Figure 18: The dot product between the query state and key states stabilizes for most positions, except for the initial and recent positions. The X-axis shows the positions of key states on a logarithmic scale.

where $K_{C_i} = [k_{C_i,1},...,k_{C_i,l_{C_i}}], V_{C_i} = [v_{C_i,1},...,v_{C_i,l_{C_i}}], A_{C_i} = [\exp\frac{Qk_{C_i,1}^\top}{\sqrt{d}},...,\exp\frac{Qk_{C_i,l_{C_i}}^\top}{\sqrt{d}}], A = [\exp\frac{Qk_1^\top}{\sqrt{d}},...,\exp\frac{Qk_l^\top}{\sqrt{d}}], a_{C_i,j} = \exp\frac{Qk_{C_i,j}^\top}{\sqrt{d}}$, and $a_j = \exp\frac{Qk_j^\top}{\sqrt{d}}$ are the denotations of each component.

We can rewrite this formula in a hierarchical way, first computing $V_{C_i}^h$ and $A_{C_i}^h$ for each context $C_i$:

$$V_{C_i}^h = \text{Softmax}\left(\frac{Q[k_{C_i,1}^\top,...,k_{C_i,l_{C_i}}^\top]}{\sqrt{d}}\right) \times [V_{C_i,1},...,V_{C_i,l_{C_i}}], A_{C_i}^h = \text{LogSumExp}\left(\frac{Q[k_{C_i,1}^\top,...,k_{C_i,l_{C_i}}^\top]}{\sqrt{d}}\right) \tag{8}$$

Similarly, for those non-context tokens, we have:

$$V^h = \text{Softmax}\left(\frac{Q[k_1^\top,...,k_l^\top]}{\sqrt{d}}\right) \times [V_1,...,V_l], A^h = \text{LogSumExp}\left(\frac{Q[k_1^\top,...,k_l^\top]}{\sqrt{d}}\right) \tag{9}$$

After computing the values of all components, we can combine these intermediate value states while renormalizing with the aggregated attention scores, the formula is shown below:

$$O = \text{Softmax}\left(A_{C_1}^h,...,A_{C_N}^h,A^h\right) \times [V_{C_1}^h,...,V_{C_N}^h,V^h] \tag{10}$$

### D.2 HIERARCHICAL FORMULA FOR APE.

After incorporating all components in APE, we have a new $V_{C_i}^{h'}$ and $A_{C_i}^{h'}$ for each context $C_i$:

$$V_{C_i}^{h'} = \text{Softmax}\left(\frac{Q[k_{C_i,1}^\top,...,k_{C_i,l_{C_i}}^\top]}{T\cdot\sqrt{d}}\right) \times [V_{C_i,1},...,V_{C_i,l_{C_i}}], A_{C_i}^{h'} = S\cdot\text{LogSumExp}\left(\frac{Q[k_{C_i,1}^\top,...,k_{C_i,l_{C_i}}^\top]}{T\cdot\sqrt{d}}\right) \tag{11}$$

For the non-context tokens, including our shared prefix, the formulas of $V^{h'}$ and $A^{h'}$ remain unchanged. Here, we introduce separate terms $V_P^{h'}$ and $A_P^{h'}$ for the shared prefix. Combining them, we have:

$$O = \text{Softmax}\left(A_P^{h'},A_{C_1}^{h'},...,A_{C_N}^{h'},A^{h'}\right) \times [V_P^{h'},V_{C_1}^{h'},...,V_{C_N}^{h'},V^{h'}] \tag{12}$$

### D.3 RELATION WITH EQUATION 5.

At last, we show that it can be rewritten as Equation 5, with the only difference being that all contexts are treated as a whole. For an token from the position $j$ in context $C_i$, the final attention score $a''_{C_i,j}$ is

$$a''_{C_i,j} = \frac{\exp(Qk_{C_i,j}^\top / T\sqrt{d})}{\sum_{n=1}^{N}\sum_{t=1}^{l_{C_n}}\exp(Qk_{C_i,t}^\top / T\sqrt{d})}$$
$$\cdot \frac{\exp\left(S \cdot \text{LogSumExp}\left(\frac{Q[k_{C_1,1}^\top,...,k_{C_1,l_{C_1}}^\top,...,k_{C_n,1}^\top,...,k_{C_n,l_{C_n}}^\top]}{T \cdot \sqrt{d}}\right)\right)}{\exp\left(S \cdot \text{LogSumExp}\left(\frac{Q[k_{C_1,1}^\top,...,k_{C_1,l_{C_1}}^\top,...,k_{C_n,1}^\top,...,k_{C_n,l_{C_n}}^\top]}{T \cdot \sqrt{d}}\right)\right) + \exp\left(\text{LogSumExp}\left(\frac{Q[k_1^\top,...,k_l^\top]}{T \cdot \sqrt{d}}\right)\right)}$$
$$(13)$$

$$= \frac{\exp(Qk_{C_i,j}^\top / T\sqrt{d})}{\sum_{n=1}^{N}\sum_{t=1}^{l_{C_n}}\exp(Qk_{C_n,t}^\top / T\sqrt{d})} \cdot \frac{(\sum_{n=1}^{N}\sum_{t=1}^{l_{C_n}}\exp(Qk_{C_n,t}^\top / T\sqrt{d}))^S}{\sum_{n=1}^{N}(\sum_{t=1}^{l_{C_n}}\exp(Qk_{C_n,t}^\top / T\sqrt{d}))^S + \sum_{t=1}^{l}\exp(Qk_t^\top / \sqrt{d})}$$
$$(14)$$

$$= \frac{\exp(Qk_{C_i,j}^\top / T\sqrt{d}) \cdot (\sum_{t=1}^{l_{C_i}}\exp(Qk_{C_i,t}^\top / T\sqrt{d}))^{(S-1)}}{(\sum_{n=1}^{N}\sum_{t=1}^{l_{C_n}}\exp(Qk_{C_n,t}^\top / T\sqrt{d}))^S + \sum_{t=1}^{l}\exp(Qk_t^\top / \sqrt{d})} = \frac{a'_{C_i,j}}{(\sum_{n=1}^{N}\sum_{t=1}^{l_{C_n}}a'_{C_i,t})^S + \sum_{t=1}^{l}a_t}$$
$$(15)$$

This formula is equivalent to Equation 5, except it combines the prefix and other non-context tokens (i.e., query and generated tokens) for simplicity. For an token from position $j$, we can derive $a''_j$ as

$$a''_j = \frac{\exp(Qk_j^\top / \sqrt{d})}{\sum_{n=1}^{N}(\sum_{t=1}^{l_{C_n}}\exp(Qk_{C_n,t}^\top / T\sqrt{d}))^S + \sum_{t=1}^{l}\exp(Qk_t^\top / \sqrt{d})} = \frac{a_j}{(\sum_{n=1}^{N}\sum_{t=1}^{l_{C_n}}a'_{C_i,t})^S + \sum_{t=1}^{l}a_t}$$
$$(16)$$

Combining these two components, we obtain the final formula presented in Equation 5.

### D.4 EFFICIENT IMPLEMENTATION.

To combine the computation for context and non-context tokens, we employ flash attention twice—once for each part—and then merge these results. This only introduces a marginal computational overhead.

```
def ape_attention(query, key, value, temperature, scale):
    # split key and value states into context and non-context parts
    key_context, key_other = key
    value_context, value_other = value
    logits_context, lse_context = flash_attn(query, key, value, temperature)
    logits_other, lse_other = flash_attn(query, key, value)
    lse_context = lse_context * scale
    attn_weights = [lse_context, lse_other]
    attn_weights = Softmax(attn_weights)
    value_states = [logits_context, logits_other]
    attn_output = attn_weights @ value_states
```

### D.5 FUTURE DIRECTIONS.

The hierarchical formulation of APE can naturally extend to more complex tree structures, as illustrated in Figure 19. This flexibility allows each user query to be enriched with external knowledge organized in such structures, demonstrating the capability of APE in handling structured external data effectively.

## E COMPARING APE WITH LONG-CONTEXT LLMS.

In Table 7, we further compare APE with Long-context LLM, including: (i) *Prompt Compression*: Truncation, LLMLingua2 (Pan et al., 2024), (ii) *KV Cache Eviction*: StreamingLLM (Xiao et al., 2023), (iii) *Long-context FT*: Llama-3-8B-Instruct-262K (Gradient, 2024), Llama-2-7B-Instruct-32K (Together, 2023), (iv) *length extrapolation*: Self-Extend (Jin et al., 2024). Experimental results showcase that APE consistently outperforms all existing long-context LLM methods. We hypothesize

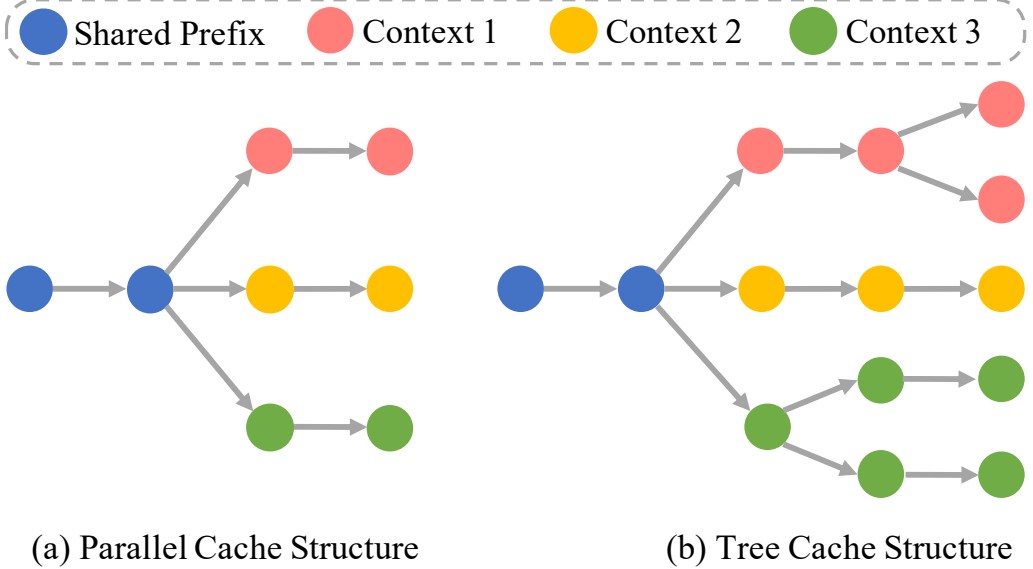

(a) Parallel Cache Structure                    (b) Tree Cache Structure

Figure 19: Beyond the parallel cache structure in our paper, APE can handle more complex cache structures, where each context forms a tree-like hierarchy. In this setup, computations can be performed hierarchically along each branch, progressively merging intermediate results into the final value state.

Table 7: Comparison between APE and long-context LLMs on the LongBench (Bai et al., 2023).

| Method | NarratQA | Qasper | MultiFQA | GovReport | QMSum | LCC |
|---|---|---|---|---|---|---|
| LLaMA-3-8B-Instruct | 19.32 | 32.83 | 43.38 | 27.89 | 22.40 | 53.22 |
| LLMLingua2 | 21.00 | 25.78 | 48.92 | 27.09 | 22.34 | 16.41 |
| StreamingLLM | 16.99 | 28.94 | 11.99 | 25.65 | 19.91 | 40.02 |
| Long-context FT | 14.88 | 21.70 | 47.79 | **32.65** | **24.76** | 55.12 |
| Self-Extend | 24.82 | 37.94 | 50.99 | 30.48 | 23.36 | 58.01 |
| +APE | **26.87** | **39.14** | **59.12** | 29.10 | 23.08 | **66.09** |

| Method | RepoBench-P | HotpotQA | 2WikiMQA | MuSiQue | MultiNews | Average |
|---|---|---|---|---|---|---|
| LLaMA-3-8B-Instruct | 38.15 | 44.24 | 21.01 | 20.47 | 23.63 | 31.50 |
| LLMLingua2 | 20.56 | 40.16 | 24.72 | 20.85 | 21.34 | 26.29 |
| StreamingLLM | 26.16 | 32.76 | 20.12 | 17.32 | 21.49 | 23.76 |
| Long-context FT | 43.05 | 15.89 | 10.49 | 8.74 | 24.28 | 27.21 |
| Self-Extend | 41.83 | **51.09** | 24.17 | **28.73** | **24.11** | 35.96 |
| +APE | **49.43** | 50.11 | **28.06** | 25.79 | 22.40 | **38.11** |

that this improvement stems from APE enabling queries to access all past contexts, thereby enhancing LLM's retrieval ability. However, due to the limitations of APE in identifying relationships between contexts, we do not emphasize its performance on current long-context benchmarks in our main paper.

## F APE CACHE VERSUS PREFIX CACHE

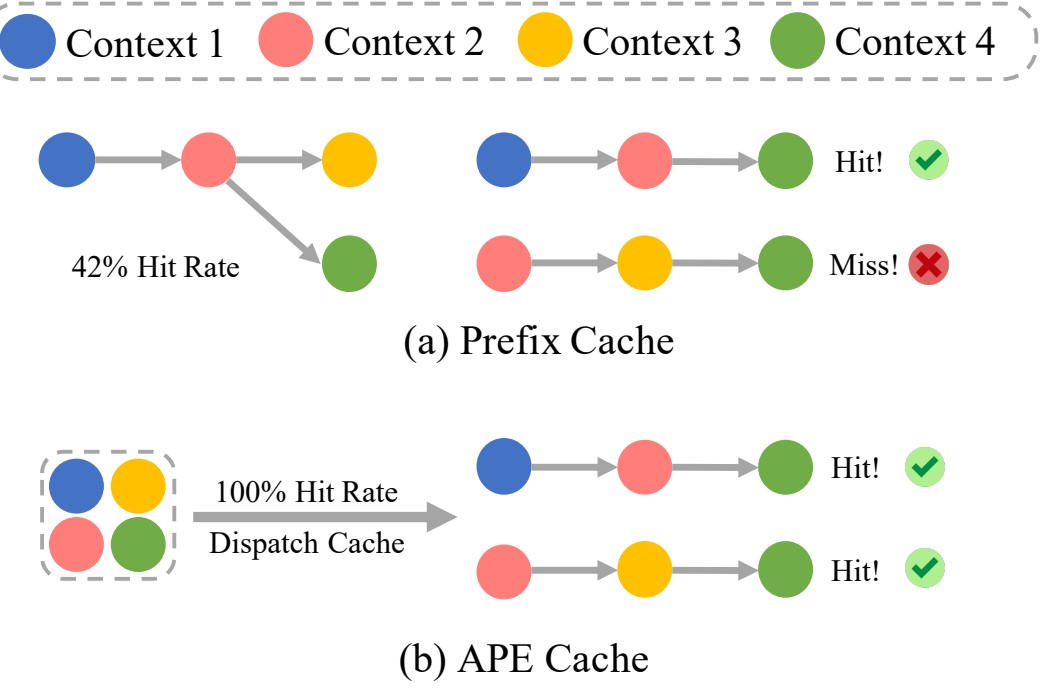

(a) Prefix Cache

(b) APE Cache

Figure 20: APE cache can keep a 100% hit rate while the Prefix cache only has a 42% hit rate.

Finally, we compare our APE cache with the prefix cache to demonstrate its advantages when serving multiple queries in the CAG setting. Figure 20 illustrates a scenario with four contexts where both caching strategies are allocated identical budgets. Each query retrieves three contexts. Under these conditions, the prefix cache can only match a limited number of combinations, resulting in an average hit rate of 41.7%, whereas the APE cache consistently achieves a 100% hit rate. This performance gap becomes more pronounced as the number of contexts increases, resulting in many more combinations.

