# OpenReview forum: "APE: Faster and Longer Context-Augmented Generation via Adaptive Parallel Encoding"
_ICLR.cc/2025/Conference — ICLR 2025 Poster_

### Official Review · Reviewer_kKET · 2024-10-28

**Soundness:** 4
**Presentation:** 2
**Contribution:** 3
**Rating:** 6
**Confidence:** 4

**Summary:**

APE proposed the adaptive parallel encoding of LLM prompt, to speedup the prefill stage without limited by pretrained context length. APE add sink tokens (shared prefix) and split the context into chunks and perform attention inside of it. During decoding stage, it attends to every previous KV without adjusting the RoPE, so it will not exceed the pretrained context length.

**Strengths:**

- Nice empirical analysis to show value state can be merged & key states are similar for each context chunk.
- Speeding up prefill stage drastically.

**Weaknesses:**

1. The methodology might be too similar, with fixed step sparse attention with sink token. When we draw the attention mask of APE, the attention mask itself is extremely similar to the step attention mechanism. However, the difference between APE and APE is that APE reuses the RoPE embeddings rather than just extends them. However, treating the RoPE index is the same as streaming with fixed step sparse attention. Therefore, I think the scientific contribution of methodology is limited.

2. The performance of the method is mostly proved by empirical results. I do not think these empirical results are weaknesses, but I have concerns about the lack of important comparisons with previous technologies to extend the context window and speed up prefill.

- Lack of comparison with training-free context extension method (Self-Extend: https://github.com/datamllab/LongLM).

- Lack of comparing with pretrained long context LLM. This might not be a big problem, but we need to know what performance is the upper limit if the LLM is already trained in a long context. The LLMs used for experiments are all short context (and maybe a little bit old at this point) models. I am concerned that using a long-context model such as Qwen2 or Llama3.1 that supports 128k tokens with a large-scale GPU cluster might lead to better performance than APE.

- Lack of comparing with techniques to speed up the prefill stage.

--- Some minor improvements ---

- Figure 3 and 4 are quite hard to understand. Can you increase the font size? And also I do not think we need visualized every layers. Can you randomly sample some layers and show them only? (e.g., range(0, 32, 4)).
- In Table 1, there is a typo in Gemma2. Maybe you shifted the columns to the left.
- Tables 2, 4, and 5 might have a too large a size.
- Table 2 should improve the formatting.

**Questions:**

1. In efficiency analysis, the claim `976x faster` might be a little bit overclaiming because there is no performance (accuracy) evaluation on 512k. Can you add relevant benchmarks? (InfiniteBench: https://github.com/OpenBMB/InfiniteBench/, LOFT: https://github.com/google-deepmind/loft, RULER: https://github.com/NVIDIA/RULER)

2. Table 5: what is the average context length of APE and real-world, end-to-end latency, including retrieval latency? As far as I understand the CRAG paper, the context length is quite limited with a smaller size (2k~4k) due to the speed of retrieval.

3. What do you mean that `query and generation lengths were fixed at 256 tokens` in line 464? Is that mean, the chunk size of APE is 256? As far as I see the latency, I think 256 means the chunk size, which is NOT used in performance evaluation. Do you have any performance evaluation with 256 chunk size? If my understanding is correct, then the reported latency is might be significantly different with real-world scenario.

---

> ### Author Response · Authors · 2024-12-03
> **Response to Reviewer kKET (1/4)**
>
> Thank you so much for the insightful and valuable comments! They are very helpful for further improving the clarity and quality of our paper. We'll revise our manuscript in the next version to address all of your concerns.
>
> ## **Q1**: The methodology might be too similar, with fixed step sparse attention with sink token. When we draw the attention mask of APE, the attention mask itself is extremely similar to the step attention mechanism. However, the difference between APE and APE is that APE reuses the RoPE embeddings rather than just extends them. However, treating the RoPE index is the same as streaming with fixed step sparse attention. Therefore, I think the scientific contribution of methodology is limited.
>
> **A1**: Our APE is designed specifically for context-augmented generation, where multiple contexts are stored in an external database. The attention mask in APE has two distinctive characteristics that set it apart. First, in the context processing phase, each context can be encoded independently without attending to other contexts. This design enables pre-caching of context KV cache, improving computational efficiency. Second, when inputting the query, it can attend to all contexts, ensuring access to information across the entire context length.
>
> These architectural choices fundamentally differentiate APE from previous methods that rely on fixed-step sparse attention with sink tokens. Such earlier approaches face two critical limitations: they require computationally expensive real-time context re-encoding, and their restricted attention masks prevent queries from effectively accessing the complete context, resulting in suboptimal performance.
>
> The primary novelty of APE's attention mask lies in its ability to enable a new setting for faster and longer context-augmented generation, while maintaining a good performance. APE represents the first method to successfully achieve these in context-augmented generation, marking a significant advancement in the field.
>
> ---
>
> ## **Q2**: Lack of comparison with training-free context extension method (Self-Extend: https://github.com/datamllab/LongLM).
>
> **A2**:  First, we need to clarify that APE is primarily designed for context-augmented generation, while the long-context serves as a by-product. Here, we added more baselines for experimental results on the LongBench dataset [1], where we evaluated 11 subsets in total and compared APE with Self-Extend [2]. The results in Table R1 show that APE achieves a 6.61% and 2.15%improvement over the base model and Self-Extend, respectively. Therefore, APE can serve as a SOTA method for length extension.
>
> Table R1: Comparison of APE and Self-Extend on the LongBench dataset.
>
> | Method              | NarratQA | Qasper | MultiFQA | GovReport | QMSum | LCC   | RepoBench-P | HotpotQA | 2WikiMQA | MuSiQue | MultiNews | Average |
> | ---- | --- | --- | -------- | --------- | ----- | ----- | ----------- | -------- | -------- | ------- | --------- | ------- |
> | LLaMA-3-8B-Instruct | 19.32    | 32.83  | 43.38    | 27.89     | 22.40 | 53.22 | 38.15       | 44.24    | 21.01    | 20.47   | 23.63     | 31.50   |
> | Self-Extend         | 24.82    | 37.94  | 50.99    | 30.48     | 23.36 | 58.01 | 41.83       | 51.09    | 24.17    | 28.73   | 24.11     | 35.96   |
> | APE                 | 26.87    | 39.14  | 59.12    | 29.10     | 23.08 | 66.09 | 49.43       | 50.11    | 28.06    | 25.79   | 22.40     | 38.11   |

---

> ### Author Response · Authors · 2024-12-03
> **Response to Reviewer kKET (2/4)**
>
> ## **Q3**: Lack of comparing with pre-trained long context LLM. This might not be a big problem, but we need to know what performance is the upper limit if the LLM is already trained in a long context. The LLMs used for experiments are all short context (and maybe a little bit old at this point) models. I am concerned that using a long-context model such as Qwen2 or Llama3.1 that supports 128k tokens with a large-scale GPU cluster might lead to better performance than APE.
>
> **A3**: First, we need to clarify that APE is primarily designed for context-augmented generation, while the long-context serves as a by-product. Here, we compared the performance of APE and sequential encoding using the LLaMA-3.1-8B-Instruct model as the pre-trained long context LLM on the LongBench dataset. Our evaluation in Table R2 revealed that APE exhibited only a marginal performance decrease of 1.67% compared to Sequential Encoding. This minimal difference does not diminish APE's advantages, as achieving the slight performance gain through sequential encoding would require substantial computational overhead during prefilling.  Additionally, starting from a pre-trained long context LLM, APE can always further extend to much longer contexts and lead to better performance.
>
> ---
>
> ## **Q4**: Lack of comparison with techniques to speed up the prefill stage.
>
> **A4**:  Thank you for your suggestion. We additionally add MInference [3] as another baseline for prefilling acceleration. As shown in Table R3, APE achieves up to 28x acceleration in prefilling compared to Sequential Encoding with MInference for 128K-token contexts. Furthermore, APE's prefilling overhead scales linearly, consuming less than 10% of the total inference time, while baseline methods require over 50% as context length increases. APE also demonstrates superior versatility across various settings, whereas MInference introduces additional overhead and slower inference speeds when processing short contexts and large batches.
>
> Table R3: Latency on H100 GPU: [prefilling / total time (ms)]. We compared different strategies with various context lengths and batch sizes.
>
> |         Context Length          | 4K        | 8K        | 16K        | 32K        | 64K         | 128K        |
> | :-----------------------------: | :-------- | :-------- | :--------- | :--------- | :---------- | :---------- |
> |       Sequential (bsz=1)        | 132/2383  | 276/2558  | 635/3204   | 1646/4713  | 4892/8804   | 16635/22252 |
> | Sequential + MInference (bsz=1) | 964/3221  | 1297/3609 | 1623/4201  | 2206/5274  | 3563/7458   | 6457/12085  |
> |        APE (bsz=1)         | 28/2279   | 40/2322   | 68/2637    | 122/3189   | 226/4138    | 421/6038    |
> |       Sequential (bsz=4)        | 523/3129  | 1084/4179 | 2558/6474  | 6599/12055 | 19468/28367 | 65928/83869 |
> | Sequential + MInference (bsz=4) | 3789/6373 | 4995/8099 | 6300/10244 | 8470/13944 | 13462/22313 | 24199/42013 |
> |        APE (bsz=4)         | 69/2675   | 94/3189   | 150/4066   | 298/5754   | 430/9329    | 858/18799   |

---

> ### Author Response · Authors · 2024-12-03
> **Response to Reviewer kKET (3/4)**
>
> ## **Q5**: In efficiency analysis, the claim 976x faster might be a little bit overclaiming because there is no performance (accuracy) evaluation on 512k. Can you add relevant benchmarks? (InfiniteBench: https://github.com/OpenBMB/InfiniteBench/, LOFT: https://github.com/google-deepmind/loft, RULER: https://github.com/NVIDIA/RULER)
>
> **A5**: First, we need to clarify that APE is primarily designed for context-augmented generation. However, as existing RAG benchmarks do not fully support 128K context lengths, we demonstrated APE's effectiveness by evaluating it on the RULER [4] and Needle In A Haystack [5] datasets with 128K context lengths, where the original context is divided into 8 chunks of 16K positions. As shown in Table R4, APE demonstrates only a 5% performance decrease compared to sequential encoding. This decline is primarily observed in tasks requiring inter-context dependencies, specifically in VT and FWE subtasks. Compared to other baselines, APE even outperforms Quest [6] and approaches ShadowKV [7] that uses 128K positions with dynamic sparse attention. Therefore, APE should work for longer contexts, especially context-augmented generation without inter-context dependencies. These results, combined with the latency analysis, show the effectiveness and efficiency of APE for extremely long contexts.
>
> Table R4: Experiment results of APE on the RULER dataset.
>
> | Model | N-S1 | N-S2 | N-MK1 | N-MK2 | N-MQ | N-MV | QA-1 | QA-2 | VT | FWE | Average |
> |---|---|---|---|---|---|---|---|---|---|---|---|
> | LLaMA-3.1-8B-Instruct | 100 | 100 | 98.96 | 91.67 | 98.96 | 95.31 | 82.29 | 47.92 | 68.96 | 71.18 | 85.525 |
> | Quest | 100 | 98.96 | 97.92 | 34.38 | 93.49 | 88.54 | 70.83 | 44.79 | 65.63 | 68.4 | 76.294 |
> | ShadowKV | 100 | 100 | 100 | 83.33 | 97.92 | 92.19 | 81.25 | 48.96 | 67.08 | 64.93 | 83.566 |
> | APE | 100 | 96.88 | 97.92 | 85.42 | 97.92 | 93.75 | 75 | 46.88 | 56.04 | 59.72 | 80.953 |
>
> In Table R5, we further validate that APE can always retrieve the needle from the 128K context on the Needle In A Haystack dataset.
>
> Table R5: Experiment results of APE on the Needle In A Haystack dataset.
>
> |                      | 1K   | 10K  | 19K  | 28K  | 37K  | 46K  | 55K  | 64K  | 73K  | 82K  | 91K  | 100K | 109K | 118K | 128K |
> | -------------------- | ---- | ---- | ---- | ---- | ---- | ---- | ---- | ---- | ---- | ---- | ---- | ---- | ---- | ---- | ---- |
> | Average Accuracy (%) | 100  | 100  | 100  | 100  | 100  | 100  | 100  | 100  | 100  | 100  | 100  | 100  | 100  | 100  | 100  |
>
> Additionally, in the updated efficiency analysis Table R3, we only evaluate from 4K to 128K, which matches the maximum length in the above benchmarks. Therefore, the current performance and efficiency results should be consistent.
>
>
> ---
>
> ## **Q6**: Table 5: what is the average context length of APE and real-world, end-to-end latency, including retrieval latency? As far as I understand the CRAG paper, the context length is quite limited with a smaller size (2k~4k) due to the speed of retrieval.
>
> **A6**: In our implementation of RAG on the CRAG benchmark, we employ sequential encoding to retrieve the top 10 chunks, each approximately 200 tokens in length. In contrast, when using APE, we retrieve the same number of chunks (top 10), but with significantly larger chunks of approximately 2,000 tokens each. Therefore, this longer context led to the improved performance in Table 5. Notably, despite the larger chunk size, APE demonstrates better latency as shown in Table R6, with improvements in both retrieval and prefilling time.
> Therefore, the retrieval time is not a bottleneck on this benchmark. This efficiency gain occurs because the increased chunk length reduces the size of the database that needs to be processed, thereby accelerating retrieval.
>
> Table R6: End-to-end latency of APE and sequential encoding on the CRAG benchmark.
>
> | Method              | Retrieval Time (ms) | Prefilling Time (ms) | Generation Time (ms) | Total Time (ms) |
> | ---- | --- | ----| -------------------- | --------------- |
> | Sequential Encoding | 356  | 152                  | 632                  | 1140            |
> | APE                 | 263   | 86                   | 705                  | 1054            |
>
> While the speedup was not significant on CRAG due to its short context length, we conducted additional measurements using the Ruler benchmark to demonstrate acceleration in long-context scenarios. As shown in Table R7, with a 128K context length, prefilling becomes a significant bottleneck for sequential encoding. In this setting, APE achieved a 15 times speedup in inference time.
>
> Table R7: End-to-end latency of APE and sequential encoding on the Ruler benchmark.
>
> | Method | Prefilling Time (ms) | Generation Time (ms) | Total Time (ms) |
> | -------- | -- | ---- | ------- |
> | Sequential Encoding | 18127                  | 812                 | 18427            |
> | APE                 | 421                   | 833                  | 1254            |

---

> ### Author Response · Authors · 2024-12-03
> **Response to Reviewer kKET (4/4)**
>
> ## **Q7**: What do you mean that query and generation lengths were fixed at 256 tokens in line 464? Is that mean, the chunk size of APE is 256? As far as I see the latency, I think 256 means the chunk size, which is NOT used in performance evaluation. Do you have any performance evaluation with 256 chunk size? If my understanding is correct, then the reported latency is might be significantly different with real-world scenario.
>
> **A7**: Here 256 doesn't mean the chunk size. Instead, it means that our query (w/o context) should be no more than 256 tokens, while the model will generate at most 256 new tokens. With the additional context length from 4K to 512K tokens, this setting simulates the real-world scenarios with long context, short query, and short response, as shown in Table 1 of the original paper. For the context part, it may be a very long context, concatenated by multiple chunks. However, the size of each chunk should approach the model's maximum sequence length (e.g., 8K tokens for the LLaMA-3-8B-Instruct model).
>
> ---
>
> **References**:
>
> [1] Bai, Yushi, et al. "Longbench: A bilingual, multitask benchmark for long context understanding." arXiv preprint arXiv:2308.14508 (2023).
>
> [2] Jin, Hongye, et al. "Llm maybe longlm: Self-extend llm context window without tuning." arXiv preprint arXiv:2401.01325 (2024).
>
> [3] Jiang, Huiqiang, et al. "Minference 1.0: Accelerating pre-filling for long-context llms via dynamic sparse attention." arXiv preprint arXiv:2407.02490 (2024).
>
> [4] Hsieh, Cheng-Ping, et al. "RULER: What's the Real Context Size of Your Long-Context Language Models?." arXiv preprint arXiv:2404.06654 (2024).
>
> [5] Greg Kamradt. Needle in a haystack - pressure testing llms. 2023.
>
> [6] Tang, Jiaming, et al. "Quest: Query-Aware Sparsity for Efficient Long-Context LLM Inference." arXiv preprint arXiv:2406.10774 (2024).
>
> [7] Sun, Hanshi, et al. "ShadowKV: KV Cache in Shadows for High-Throughput Long-Context LLM Inference." arXiv preprint arXiv:2410.21465 (2024).

---

> ### Comment · Reviewer_kKET · 2024-12-03
> **Reviewer Response to Author's Rebuttal**
>
> Thank you for insightful and extensively verified rebuttal. The response clearly resolve my all concerns. I think I am misunderstand the concept of the whole paper.
>
> ---
>
> However, I think the author's claim "as existing RAG benchmarks do not fully support 128K context length" is not true, because the LOFT benchmark already supports RAG benchmark up to 1M context document. I think LOFT benchmark is the most important for APE. The LOFT is perfect benchmark to test APE, because (1) the context is concat of thousands of short documents, which is APE targeted. (2) The prompt is shared across queries, therefore partial prefix caching that emphasized by authors. (3) The context length is extremely long (1M) therefore the prefill stage is extremely important to be speedup.
>
> I wonder we can compare the performance in LOFT (but I understand we do not have time now...). As far as, I played with that dataset, 128k evaluation on RAG-NQ dataset only took 10 to 20 minutes if the prompt is cached properly. I hope we can see updated result if possible during rebuttal, or final revision.
>
> ---
>
> I check the provided code, and it is little bit disappointing the implementation is not very efficient. I was expected the codebase is based on PagedAttention framework with prefix caching features such as SGLang. I think the efficiency of implementation is extremely important as the paper proposing efficient mechanism. I want to encourage the author can contribute the open-sources such as vLLM and SGlang with APE in future time after the submission.
>
> ---
>
> After re-considering main manuscript with updated rebuttal, I want to raise my score from 3 (reject) to 6 (borderline accept). Again, thank you for extensively verified rebuttal, and pay respect to authors for trying the best until last hours.

---

> > ### Author Response · Authors · 2024-12-03
> >
> > Dear Reviewer kKET,
> >
> > Thank you very much for the increased score and for recommending the LOFT dataset. We will conduct additional experiments using LOFT subsets and include these results in our rebuttal. If time constraints prevent us from completing these experiments before the rebuttal deadline, we commit to providing comprehensive results in the final revision.
> >
> > Thanks,
> >
> > The Authors

---

> > ### Author Response · Authors · 2024-12-03
> > **Future Plan for Efficient Implementation**
> >
> > Thank you agin for your encouragement and suggestions! We strongly agree that prefix caching is very useful for speeding up computations, and APE enables systems to generalize the prefix caching mechanisms particularly when multiple contexts are available. **Therefore, we plan to reach out to vLLM, SGLang teams, and bring APE to these frameworks.**
> >
> > Additionally, we want to further discuss about the potential of APE in improving prefix caching mechanisms. For example, normal prefix cache requires to recompute prefix sequence [c0, c1], [c1, c2], [c0, c2] three times and bring exponential explosion when multiple contexts are combined. In comparison, APE allows us to cache c0, c1, c2 independently. Therefore, we can directly load all potential combinations of {c0, c1, c2} with only linear complexity in memory, bringing potential for further memory saving and speedup. As a result, our approach complements and can enhance previous prefix caching mechanisms.

---

> ### Author Response · Authors · 2024-12-04
> **Additional Experiments on the LOFT benchmark**
>
> Based on your suggestions, we further evaluated our approach on four text retrieval tasks from the LOFT benchmark using LLaMA-3.1-8B-Instruct as our base model, with performance measured by the F1 score between ground truth and model predictions. To simulate realistic scenarios where RAG systems may retrieve suboptimal content, we randomly sampled passages from the corpus as context, enabling analysis of how performance scales with additional context integration (Otherwise, we only need a short-context LLM to achieve a good performance if the retriever is very accurate, which is not the setting that we target on). For the 128k version of LOFT, we compared APE against Sequential Encoding using varying context lengths in Table R8.
>
> Table R8. Comparison of APE and Sequential Encoding on four text retrieval tasks from the 128k version of LOFT benchmark using the LLaMA-3.1-8B-Instruct model.
>
> |                   | Inference Time | Arguana | Fever | NQ    | SciFact | Average |
> | :------------------------------ | :-------- | :--------- | :--------- | :---------- | :---------- | ------------------------------- |
> | 8K, Sequential | 847            | 11.15   | 7.78 | 17.78 | 7.74 | 11.11 |
> | 16K, Sequential | 1277           | 11.20   | 9.78 | 17.81 | 9.49 | 12.07 |
> | 32K, Sequential | 2413 | 12.15 | 10.22 | 15.78 | 9.74 | 11.97 |
> | 64K, Sequential | 5870 | 15.34 | 13.12 | 19.64 | 16.12 | 16.06 |
> | 128K, Sequential | 18039 | 12.84 | 14.19 | 24.54 | 16.88 | 17.11 |
> | 128K, APE | 1825 | 16.32 | 14.70 | 21.91 | 15.72 | 17.16 |
>
> Our analysis yielded several key findings:
>
> * APE demonstrates superior performance on average compared to all baselines, validating APE’s scalability in processing hundreds/thousands of retrieved texts in parallel.
>
> * APE achieves performance comparable to the 128K Sequential Encoding baseline while utilizing the same context. Although transferring from sequential encoding to APE would typically be expected to cause performance degradation, we hypothesize that APE recovers this by effectively shortening the distance between the query and all examples. This mechanism helps address the “lost in the middle” problem common in long-context LLMs.
>
> * APE demonstrates remarkable computational efficiency, with its 128K variant requiring inference times comparable to Sequential Encoding at much shorter context lengths (between 16K and 32K), while achieving approximately 10 times faster processing compared to Sequential Encoding at equivalent context length.
>
> We also compared APE with Sequential Encoding using the LLaMA-3.1-8B-Instruct model on the 1m version of LOFT, as shown in Table R9. While APE improves 5.26% on average using 8$\times$ more retrieved texts, it only requires 1/3 inference time by pre-caching these texts.
>
> Table R9. Comparison of APE and Sequential Encoding on four text retrieval tasks from the 1m version of LOFT benchmark using the LLaMA-3.1-8B-Instruct model.
>
> |                   | Inference Time | Arguana | Fever | NQ    | SciFact | Average |
> | :------------------------------ | :-------- | :--------- | :--------- | :---------- | :---------- | ------------------------------- |
> | 128K, Sequential | 18039          | 11.28 | 8.13 | 17.55 | 7.54 | 11.13 |
> | 1M, APE          | 5073           | 15.89 | 15.13 | 20.78 | 14.55 | 16.59 |

---

### Official Review · Reviewer_onHd · 2024-11-02

**Soundness:** 3
**Presentation:** 3
**Contribution:** 3
**Rating:** 6
**Confidence:** 4

**Summary:**

This paper introduces Adaptive Parallel Encoding, a method to improve the efficiency of large language models when processing multiple external contexts. APE pre-caches key-value states of contexts separately and enables position reuse during inference. The method consists of three key components: a shared prefix to align initial token distributions, a scaling factor to offset increased attention weights, and lower attention temperature to focus on semantically important tokens. The authors demonstrate that APE achieves a 976× speedup for long context generation while maintaining 93% accuracy compared to sequential encoding.

**Strengths:**

1. The authors provide thorough empirical analysis to understand the behavior of attention mechanisms and KV states.
2. The method is practical as it requires no additional training and can be implemented with minimal modifications to existing architectures.
3. Comprehensive evaluation across multiple tasks (ICL, RAG, long-context understanding)

**Weaknesses:**

1. The performance evaluation is primarily focused on 8k context length, which feels insufficient given that many open-source LLMs now support context lengths of 128k or more. This restricted evaluation scope makes it difficult to assess the method's scalability to longer contexts.
2. Compared to sequential encoding, APE introduces a non-negligible performance degradation at the 8k length scale, raising concerns about its effectiveness at longer context lengths.

**Questions:**

1. How does the computational complexity of APE scale with increasing context length compared to sequential encoding?
2. Why does the method perform particularly poorly on code completion tasks with continuous long contexts?
3. How sensitive is the method to the choice of hyperparameters (scaling factor and attention temperature) across different models and tasks?

---

> ### Author Response · Authors · 2024-12-03
> **Response to Reviewer onHd (1/2)**
>
> Thank you for your valuable comments and suggestions. We will carefully revise our paper based on your comments. Our responses to your questions are detailed below. We would greatly appreciate your input on whether our revisions address your concern
>
> ---
>
> ## **Q1**: The performance evaluation is primarily focused on 8k context length, which feels insufficient given that many open-source LLMs now support context lengths of 128k or more. This restricted evaluation scope makes it difficult to assess the method's scalability to longer contexts.
>
> **A1**:  First, we need to clarify that APE is primarily designed for context-augmented generation. However, as existing RAG benchmarks do not fully support 128K context lengths, we demonstrated APE's effectiveness by evaluating it on the RULER [1] and Needle In A Haystack [2] datasets with 128K context lengths, where the original context is divided into 8 chunks of 16K positions. We also include Quest [3] and ShadowKV [4] as two baselines using dynamic sparse attention and 128K positions. As shown in Table R2, APE demonstrates only a 5% performance decrease compared to Sequential Encoding. This decline is primarily observed in tasks requiring inter-context dependencies, specifically in VT and FWE subtasks. Compared to other baselines, APE even outperforms Quest and approaches ShadowKV despite using only 1/8 positions. Therefore, APE should work for longer contexts, especially context-augmented generation without inter-context dependencies.
>
> Table R1: Experiment results of APE on the RULER dataset.
>
> | Model | N-S1 | N-S2 | N-MK1 | N-MK2 | N-MQ | N-MV | QA-1 | QA-2 | VT | FWE | Average |
> |---|---|---|---|---|---|---|---|---|---|---|---|
> | LLaMA-3.1-8B-Instruct | 100 | 100 | 98.96 | 91.67 | 98.96 | 95.31 | 82.29 | 47.92 | 68.96 | 71.18 | 85.525 |
> | Quest | 100 | 98.96 | 97.92 | 34.38 | 93.49 | 88.54 | 70.83 | 44.79 | 65.63 | 68.4 | 76.294 |
> | ShadowKV | 100 | 100 | 100 | 83.33 | 97.92 | 92.19 | 81.25 | 48.96 | 67.08 | 64.93 | 83.566 |
> | APE | 100 | 96.88 | 97.92 | 85.42 | 97.92 | 93.75 | 75 | 46.88 | 56.04 | 59.72 | 80.953 |
>
> In Table R2, we further validate that APE can always retrieve the needle from the 128K context on the Needle In A Haystack dataset.
>
> Table R2: Experiment results of APE on the Needle In A Haystack dataset.
>
> |       | 1K   | 10K  | 19K  | 28K  | 37K  | 46K  | 55K  | 64K  | 73K  | 82K  | 91K  | 100K | 109K | 118K | 128K |
> | --- | ---- | ---- | ---- | ---- | ---- | ---- | ---- | ---- | ---- | ---- | ---- | ---- | ---- | ---- | ---- |
> | Average Accuracy (%) | 100  | 100  | 100  | 100  | 100  | 100  | 100  | 100  | 100  | 100  | 100  | 100  | 100  | 100  | 100  |
>
> ---
>
> ## **Q2**: Compared to sequential encoding, APE introduces a non-negligible performance degradation at the 8k length scale, raising concerns about its effectiveness at longer context lengths.
>
> **A2**: We admit that APE will lead to performance degradation compared to sequential encoding. However, this minimal difference does not diminish APE's advantages, as achieving the slight performance gain through sequential encoding would require substantial computational overhead during prefilling. The results in Table R1 and R2 further validates this performance degradation will not increase when scaling to longer context lengths.
>
> Moreover, the reuse of positions enable APE to leverage more contexts, thereby outperforming sequential encoding in practical applications as shown in Section 6.3.
>
> ---
>
> ## **Q3**: How does the computational complexity of APE scale with increasing context length compared to sequential encoding?
>
> **A3**: We add analysis for the computational complexity of APE and Sequential Encoding. Following Section 2.1 and let the context-augmented input to be
>
> $$\mathcal{S} = {\underbrace{s_{C_0, 0}, \ldots, s_{C_0, l_0}}\_{\text{Context}}, \underbrace{s_{C_1, 0}, \ldots, s_{C_1, l_1}}\_{\text{Context 1}}, \ldots, \underbrace{s_{C_{N-1}, 0}, \ldots, s_{C_{N-1}, l_{N-1}}}\_{\text{Context N-1}}, \underbrace{s_0, \ldots, s_l}\_{\text{Query}}}, $$ where $l_0 + \ldots + l_{N-1} \gg l$, we analyze the prefilling time of Sequential Encoding and APE.
>
> * Sequential Encoding: $O((l_0+...+l_{N-1}+l)^2)$
>
> * APE with pre-caching: $O((l_0+...+l_{N-1}+l)\cdot l)$
>
> * APE without pre-caching: $O(\max(l_0^2, ..., l_{N-1}^2)+((l_0+...+l_{N-1}+l)\cdot l$
>
> Therefore, APE achieves linear complexity with respect to total context length when using pre-caching, resulting in significantly improved efficiency. Even without pre-caching, APE maintains an advantage by allowing independent prefilling of each context. In this case, its complexity scales quadratically with the maximum context length, which is more efficient than Sequential Encoding's quadratic scaling with total context length.

---

> > ### Author Response · Authors · 2024-12-03
> > **Response to Reviewer onHd (2/2)**
> >
> > ## **Q4**: Why does the method perform particularly poorly on code completion tasks with continuous long contexts?
> >
> > **A4**: This is because the code generation tasks require LLMs to generate the code after the end of context. Therefore, directly applying APE will mess up the nearby and far contexts. To address this limitation, we preserve the final 500 tokens of the provided code as the initial segment of the query. This modification significantly improves APE's performance on code generation tasks, as demonstrated by the results in Table R3.
> >
> > Table R3: Comparison of APE and other methods on long-context code generation.
> >
> > |  | LCC       | RepoBench-P |
> > | :------------ | ----- | ----- |
> > | LLaMA-3-8B-Instruct    | 53.22 | 38.15 |
> > |    + LLMLingua2     | 16.41     | 20.56       |
> > |   + StreamingLLM    | 40.02     | 26.16       |
> > |  + Long-context FT  | 55.12     | 43.05       |
> > |    +Self-Extend     | 58.01     | 41.83 |
> > | + APE | **66.09** | **49.43** |
> >
> > ---
> >
> > ## **Q5**: How sensitive is the method to the choice of hyperparameters (scaling factor and attention temperature) across different models and tasks?
> >
> > **A5**: Our APE, despite introducing three hyperparameters (shared prefix, scaling factor, and attention temperature), is not very sensitive on these hyperparameters. First, the shared prefix is only applied to tasks lacking existing prefixes (e.g., GSM8K and TriviaQA). Our experimental results in Table R4 demonstrate that introducing a shared prefix increases performance by 7.38% on average, while variations in prefix length only result in a 0.4% difference in performance. These findings indicate that while the presence of a shared prefix is beneficial, APE's performance is not significantly sensitive to the prefix length itself.
> >
> > Table R4: Ablation study of prefix length on GSM8K and TriviaQA using the LLaMA-3-8B-Instruct model.
> >
> > | Prefix Length | 0     | 10    | 20    | 40    |
> > | :-----------: | :---- | :---- | :---- | ----- |
> > |   GSM8K     | 62.17 | 72.10 | 71.87 | 71.69 |
> > |   TriviaQA    | 67.48 | 72.31 | 72.51 | 72.65 |
> >
> > For the scaling factor $s$ and attention temperature $T$, we evaluated various combinations across four QA tasks from the LongBench dataset, as shown in Table R2. While these parameters are relatively sensitive compared to the prefix length, their optimal values fall within a narrow range. In our experiments, both $s$ and $T$ were selected from {0.7, 0.8, 0.9, 1.0}, making hyperparameter optimization computationally tractable on the validation set. Notably, despite LongBench examples containing varying numbers of contexts, APE's hyperparameters demonstrated robust generalization across tasks with different context lengths, indicating low sensitivity.
> >
> > Table R5: Ablation study of  the scaling factor s and attention temperature T on four subsets from LongBench using the LLaMA-3-8B-Instruct model.
> >
> > | ($s$, $T$) | MuSiQue | NarrativeQA | HotpotQA | 2WikiMQA |
> > | :-----------: | :---- | :---- | :---- | ----- |
> > | (1.0, 1.0) | 24.13   | 26.93       | 49.71    | 43.09    |
> > | (1.0, 0.9) | 26.17   | 30.71       | 47.79    | 41.19    |
> > | (0.9, 0.9) | 24.32   | 27.93       | 49.6     | 38.21    |
> > | (0.8, 0.8) | 26.19   | 27.01       | 47.32    | 35.41    |
> >
> > ---
> >
> > **References**:
> >
> > [1] Hsieh, Cheng-Ping, et al. "RULER: What's the Real Context Size of Your Long-Context Language Models?." arXiv preprint arXiv:2404.06654 (2024).
> >
> > [2] Greg Kamradt. Needle in a haystack - pressure testing llms. 2023.
> >
> > [3] Tang, Jiaming, et al. "Quest: Query-Aware Sparsity for Efficient Long-Context LLM Inference." arXiv preprint arXiv:2406.10774 (2024).
> >
> > [4] Sun, Hanshi, et al. "ShadowKV: KV Cache in Shadows for High-Throughput Long-Context LLM Inference." arXiv preprint arXiv:2410.21465 (2024).

---

> ### Author Response · Authors · 2024-12-04
> **Additional experiments on the 128K and 1M context-length LOFT benchmark**
>
> Dear Reviewer onHd,
>
> Based on the suggestions from Reviewer kKET, we further evaluated our approach on four text retrieval tasks from the LOFT benchmark to represent long-context RAG scenarios, which further validate that APE's scalability in processing hundreds/thousands of retrieved texts with the total length of 1M in parallel. We hope these results, combined to the results on the RULER and Needle-In-A-Haystack datasets, can comprehensively show the effectiveness of APE in extreme long-context scenarios.
>
> Here, we use LLaMA-3.1-8B-Instruct as our base model, with performance measured by the F1 score between ground truth and model predictions. To simulate realistic scenarios where RAG systems may retrieve suboptimal content, we randomly sampled passages from the corpus as context, enabling analysis of how performance scales with additional context integration (Otherwise, we only need a short-context LLM to achieve a good performance if the retriever is very accurate, which is not the setting that we target on). For the 128k version of LOFT, we compared APE against Sequential Encoding using varying context lengths in Table R9.
>
> Table R9. Comparison of APE and Sequential Encoding on four text retrieval tasks from the 128k version of LOFT benchmark using the LLaMA-3.1-8B-Instruct model.
>
> |                   | Inference Time | Arguana | Fever | NQ    | SciFact | Average |
> | :------------------------------ | :-------- | :--------- | :--------- | :---------- | :---------- | ------------------------------- |
> | 8K, Sequential | 847            | 11.15   | 7.78 | 17.78 | 7.74 | 11.11 |
> | 16K, Sequential | 1277           | 11.20   | 9.78 | 17.81 | 9.49 | 12.07 |
> | 32K, Sequential | 2413 | 12.15 | 10.22 | 15.78 | 9.74 | 11.97 |
> | 64K, Sequential | 5870 | 15.34 | 13.12 | 19.64 | 16.12 | 16.06 |
> | 128K, Sequential | 18039 | 12.84 | 14.19 | 24.54 | 16.88 | 17.11 |
> | 128K, APE | 1825 | 16.32 | 14.70 | 21.91 | 15.72 | 17.16 |
>
> Our analysis yielded several key findings:
>
> * APE demonstrates superior performance on average compared to all baselines, validating APE’s scalability in processing hundreds/thousands of retrieved texts in parallel.
>
> * APE achieves performance comparable to the 128K Sequential Encoding baseline while utilizing the same context. Although transferring from sequential encoding to APE would typically be expected to cause performance degradation, we hypothesize that APE recovers this by effectively shortening the distance between the query and all examples. This mechanism helps address the “lost in the middle” problem common in long-context LLMs.
>
> * APE demonstrates remarkable computational efficiency, with its 128K variant requiring inference times comparable to Sequential Encoding at much shorter context lengths (between 16K and 32K), while achieving approximately 10 times faster processing compared to Sequential Encoding at equivalent context length.
>
> We also compared APE with Sequential Encoding using the LLaMA-3.1-8B-Instruct model on the 1m version of LOFT, as shown in Table R10. While APE improves 5.26% on average using 8$\times$ more retrieved texts, it only requires 1/3 inference time by pre-caching these texts.
>
> Table R10. Comparison of APE and Sequential Encoding on four text retrieval tasks from the 1m version of LOFT benchmark using the LLaMA-3.1-8B-Instruct model.
>
> |                   | Inference Time | Arguana | Fever | NQ    | SciFact | Average |
> | :------------------------------ | :-------- | :--------- | :--------- | :---------- | :---------- | ------------------------------- |
> | 128K, Sequential | 18039          | 11.28 | 8.13 | 17.55 | 7.54 | 11.13 |
> | 1M, APE          | 5073           | 15.89 | 15.13 | 20.78 | 14.55 | 16.59 |

---

### Official Review · Reviewer_Yenp · 2024-11-03

**Soundness:** 3
**Presentation:** 2
**Contribution:** 2
**Rating:** 8
**Confidence:** 4

**Summary:**

This paper investigates the performance degradation issue in parallel encoding and analyzes the distribution patterns of key and query vectors. This paper analysis reveals that key vector distributions remain relatively similar and query vector distributions can be combined. Based on these findings, they propose APE (Adaptive Parallel Encoding) to mitigate performance losses in LLMs during parallel encoding. Specifically, APE incorporates three key components: shared prompt prefix, position-aware dynamic scaling, and attention temperature adjustment. We evaluated APE on three LLMs (Llama-3-8B, Llama-2-7B, Gemma-2-9B) across multi-document QA and few-shot ICL tasks. Results demonstrate performance improvements compared to full attention and baseline methods, with latency benchmarks showing significant speedup over full attention.

**Strengths:**

1. The research focuses on a practical and significant challenge, particularly relevant to real-world LLM applications like RAG.
2. The paper provides insightful observations and analysis regarding the feasibility of parallel encoding.

**Weaknesses:**

1. This paper demonstrates limited evidence for performance superiority over baselines across various tasks. Specifically:
  - The evaluation is restricted to only two task categories from LongBench (multi-document QA and few-shot ICL), which is insufficient to demonstrate the method's effectiveness across diverse scenarios.
  - For multi-document QA tasks, the evaluation is conducted on a limited subset of LongBench, with additional testing only on one RAG dataset in Sec 6.3. This narrow scope of evaluation fails to provide comprehensive evidence of the method's effectiveness. And Table 5 lacks crucial baseline comparisons, making it difficult to assess the relative performance improvements.
  - The evaluation of other task types in Sec 6.2 is inadequate, missing essential baseline comparisons needed for meaningful performance assessment.

2. And this paper lacks sufficient component analysis and validation.
  - The paper lacks comprehensive ablation studies to isolate and validate the contribution of each proposed component. This makes it impossible to determine whether performance improvements stem from specific modules (e.g., attention temperature adjustment) or their combination.
  - There is insufficient analytical evidence demonstrating how the three proposed components effectively address the challenges identified in the earlier sections of the paper. The causal relationship between the proposed solutions and the observed improvements needs stronger empirical support.
  - The absence of detailed component-wise analysis makes it difficult to justify the necessity and effectiveness of each module in the proposed architecture.

**Questions:**

1. Formatting issue: In Table 1 #413 Gemma-2-9B line has miss alignment.

---

> ### Author Response · Authors · 2024-12-03
> **Response to Reviewer Yenp (1/3)**
>
> Thank you for your constructive comments and suggestions. We respond to your questions below and would appreciate it if you could let us know if our response addresses your concerns.
>
> ---
>
> ## **Q1**: For multi-document QA tasks, the evaluation is conducted on a limited subset of LongBench, with additional testing only on one RAG dataset in Sec 6.3. This narrow scope of evaluation fails to provide comprehensive evidence of the method's effectiveness.
>
> **A1**: Thank you for your comment. We have added more experiments to showcase the effectiveness of our methods in context-augmented generation.
>
> * RAG on the LongBench dataset [1]. In this setting, we implement a chunking-and-retrieval approach across eight LongBench subsets following the original paper. The method involved preprocessing long contexts into chunks and utilizing Contriever [2] to retrieve the top-k most relevant chunks. We conducted experiments using four base models: LLaMA-3-8B-Instruct, LLaMA-3.1-8B-Instruct, Mistral-7B-Instruct-v0.3, and Gemma-2-9b-it. For sequential encoding, we segmented texts into chunks of approximately 200 words. When using APE, we employed larger chunks of 4,000 words, except for Gemma-2-9b-it, which used 2,000-word chunks. For LLaMA-3.1-8B-Instruct, we limited context length to 8K tokens to simulate a typical RAG setting. We also included comparisons with PCW, a baseline parallel encoding method. As shown in Table R1, R2, R3, R4, APE leads to 5.6% and 4.5% improvement on average compared to Sequential Encoding and PCW.  These findings suggest that APE enhances RAG performance by enabling the retrieval of more and longer contexts without compromising inference speed, as pre-cached contexts can be loaded directly.
>
> Table R1: RAG experiments for APE and other methods on the LongBench dataset for the LLaMA-3-8B-Instruct model. Here C200*10 means using Contriever to retrieve the top10 chunks, with each chunk containing approximately 200 words.
>
> |                     | musique | qasper | 2wikimqa | dureader | hotpotqa | narrativeqa | multifieldqa_zh | multifieldqa_en | Average  |
> |---------|---------|------|------|-----|--------------|-----------------|-----------------|---------|---------|
> | LLaMA-3-8B-Instruct | 20.7    | 41.05  | 30.02    | 9.55     | 45.9     | 20.98       | 58.54           | 45.04           | 33.97    |
> | C200*10, Sequential | 22.63   | 40.6   | 36.46    | 17.53    | 46.25    | 18.14       | 56.13           | 52.42           | 36.27    |
> | C200*20, Sequential | 27.93   | 42.71  | 38.35    | 12.65    | 49.6     | 22.78       | 57.82           | 48.94 | 37.60    |
> | C4000*20, PCW       | 18.82   | 42.59  | 40.99    | 21.57    | 47.09    | 23.29       | 54.4            | 45.05 | 36.73  |
> | C4000*20, APE       | 26.19   | 42.32  | 44.43    | 23.13    | 49.71    | 30.71       | 55.03           | 45.41 | 39.62 |
>
> Table R2: RAG experiments for APE and other methods on the LongBench dataset for the LLaMA-3.1-8B-Instruct model.
>
> |                     | musique | qasper | 2wikimqa | dureader | hotpotqa | narrativeqa | multifieldqa_zh | multifieldqa_en | Average  |
> |---------|---------|-----------|-----------|-----------|--------------|-----------------|-----------------|---------|---------|
> | LLaMA-3-8.1B-Instruct | 22.18   | 46.81  | 40.58    | 34.61    | 43.97    | 23.08       | 61.6            | 51.89           | 38.98   |
> | C200*10, Sequential | 26.31   | 38.79  | 41.42    | 33.16    | 48.42    | 23.39       | 56.54           | 55.37           | 38.29   |
> | C200*20, Sequential | 30.62   | 42.33  | 44.39    | 33.51    | 49.97    | 23.87       | 56.87           | 55.14           | 40.22   |
> | C4000*20, PCW       | 21.23   | 41.52  | 44.87    | 31.11    | 49.47    | 19.98       | 60.9            | 51.19           | 38.44   |
> | C4000*20, APE       | 26.88   | 43.03  | 50.11    | 32.1     | 55.41    | 30.5        | 62.02           | 52.51           | 42.86   |
>
>
>
> Table R3: RAG experiments for APE and other methods on the LongBench dataset for the Mistral-7B-Instruct-v0.3 model.
>
>
> |                     | musique | qasper | 2wikimqa | dureader | hotpotqa | narrativeqa | multifieldqa_zh | multifieldqa_en | Average  |
> |---------|---------|-----------|-----------|-----------|--------------|-----------------|-----------------|---------|---------|
> | Mistral-7B-Instruct-v0.3 | 10.05   | 31.08  | 22.12    | 17.68    | 32.09    | 19.68       | 32.03           | 40.38           | 25.64   |
> | C200*10, Sequential | 12.33   | 22.8   | 23.65    | 26.86    | 30.47    | 19.01       | 40.64           | 46.26           | 27.75   |
> | C200*20, Sequential | 11.58   | 21.98  | 24.44    | 20.8     | 32.79    | 16.06       | 34.43           | 38.4            | 25.06   |
> | C4000*20, PCW       | 17.58   | 35.57  | 32.97    | 18.7     | 37.05    | 14.1        | 34.69           | 40.14           | 28.85   |
> | C4000*20, APE       | 20.3    | 36.81  | 34.37    | 21.89    | 42.33    | 20.49       | 40.2            | 44.03           | 32.55   |

---

> ### Author Response · Authors · 2024-12-03
> **Response to Reviewer Yenp (2/3)**
>
> Table R4: RAG experiments for APE and other methods on the LongBench dataset for the Gemma-2-9b-it model.
>
> |                     | musique | qasper | 2wikimqa | dureader | hotpotqa | narrativeqa | multifieldqa_zh | multifieldqa_en | Average  |
> |---------|---------|-----------|-----------|-----------|--------------|-----------------|-----------------|---------|---------|
> | Gemma-2-9b-it | 22.57   | 39.99  | 48.06    | 27.4     | 47.49    | 23.11       | 50.81           | 45.35           | 38.10   |
> | C200*5, Sequential | 28.73   | 42.06  | 51.62    | 26.94    | 47.87    | 20.49       | 48.09           | 47.38           | 39.15   |
> | C200*10, Sequential | 30.69   | 42.86  | 53.55    | 28.04    | 52.05    | 24.45       | 50.25           | 48.34           | 41.28   |
> | C2000*20, PCW      | 26.27   | 46.69  | 47.59    | 23.43    | 48.95    | 27.11       | 56.69           | 49.81           | 40.82   |
> | C2000*20, APE      | 33.38   | 47.72  | 49.49    | 28.43    | 56.62    | 30.41       | 56.52           | 50.84           | 44.18   |
>
> * RAG on the ChatRAGBench dataset [3] with multiple documents. In this setting, we employed three distinct retrievers: Dragon-multiturn [3], Contriever [2], and GTE-base [4] to retrieve the top-5 relevant contexts from five multi-document subsets of the ChatRAGBench dataset. We then evaluated the performance of Llama3-ChatQA-1.5-8B using these retrieved contexts, comparing sequential encoding against APE. Additionally, we tested APE's performance when using all available contexts. The results, presented in Table R5, demonstrate that shifting  from sequential encoding to APE results in a minimal performance degradation of no more than 1%, which we consider acceptable. Notably, this performance gap becomes even smaller with lower-quality retrievers. Furthermore, when directly processing all contexts, APE not only outperforms all sequential encoding baselines but also shows substantial improvements compared to scenarios using low-quality retrievers.
>
> Table R5: RAG experiments for APE and sequential encoding on the ChatRAGBench dataset using different retrievers.
>
> | Method                       | INSCIT | Doc2Dial | TopicCQA | Qrecc | QuAC  | Average |
> | ---------------------------- | ------ | -------- | -------- | ----- | ----- | ------- |
> | dragon-multiturn, Sequential | 25.42  | 36.27    | 36.1     | 49.01 | 35.12 | 36.38   |
> | dragon-multiturn, APE        | 23.84  | 34.93    | 33.8     | 48.7  | 34.92 | 35.24   |
> | Contriever, Sequential       | 19.97  | 23.85    | 30.49    | 46.75 | 26.57 | 29.53   |
> | Contriever, APE              | 19.88  | 23.28    | 28.84    | 46.28 | 26.8  | 29.02   |
> | GTE-base, Sequential         | 21.58  | 32.35    | 33.41    | 46.54 | 30.69 | 32.91   |
> | GTE-base, APE                | 20.85  | 30.99    | 31.92    | 45.83 | 30.35 | 31.99   |
> | All contexts, APE            | 27.22  | 36.13    | 35.72    | 49.15 | 35.7  | 36.78   |
>
> ---
>
> ## **Q2**: The evaluation of other task types in Sec 6.2 is inadequate, missing essential baseline comparisons needed for meaningful performance assessment.
>
> **A2**: We added more baselines for experimental results on the LongBench dataset, where we evaluated 11 subsets in total and compared APE with LLMLingua2 [5], StreamingLLM [6], Long-context Fine-tuning, and Self-Extend [7]. The results in Table R6 show that APE achieves a 6.61% improvement over the base model. Among other length extension methods, only Self-Extend shows performance gains, though it still lags behind APE by 2.15%.
>
> Table R6: Comparison of APE and other length extension methods on the LongBench dataset.
>
> | Method              | NarratQA | Qasper | MultiFQA | GovReport | QMSum | LCC   | RepoBench-P | HotpotQA | 2WikiMQA | MuSiQue | MultiNews | Average |
> | ------------------- | -------- | ------ | -------- | --------- | ----- | ----- | ----------- | -------- | -------- | ------- | --------- | ------- |
> | LLaMA-3-8B-Instruct | 19.32    | 32.83  | 43.38    | 27.89     | 22.40 | 53.22 | 38.15       | 44.24    | 21.01    | 20.47   | 23.63     | 31.50   |
> | LLMLingua2          | 21.00    | 25.78  | 48.92    | 27.09     | 22.34 | 16.41 | 20.56       | 40.16    | 24.72    | 20.85   | 21.34     | 26.29   |
> | StreamingLLM        | 16.99    | 28.94  | 11.99    | 25.65     | 19.91 | 40.02 | 26.16       | 32.76    | 20.12    | 17.32   | 21.49     | 23.76   |
> | Long-context FT     | 14.88    | 21.70  | 47.79    | 32.65     | 24.76 | 55.12 | 43.05       | 15.89    | 10.49    | 8.74    | 24.28     | 27.21   |
> | Self-Extend         | 24.82    | 37.94  | 50.99    | 30.48     | 23.36 | 58.01 | 41.83       | 51.09    | 24.17    | 28.73   | 24.11     | 35.96   |
> | APE                 | 26.87    | 39.14  | 59.12    | 29.10     | 23.08 | 66.09 | 49.43       | 50.11    | 28.06    | 25.79   | 22.40     | 38.11   |

---

> ### Author Response · Authors · 2024-12-03
> **Response to Reviewer Yenp (3/3)**
>
> ## **Q3**: And Table 5 lacks crucial baseline comparisons, making it difficult to assess the relative performance improvements.
>
> **A3**:  Thank you for your suggestions. In Table 5, we evaluate RAG with Sequential Encoding and APE for two different external databases, with additional comparison to LLM-only generation. In these settings, APE consistently leads to performance improvement. Therefore, Table 5 only serves as one real-world RAG application for APE, where we only compares it with Sequential Encoding.
>
> To compare with more baseline, our evaluation in Tables R1-R6, which includes various methods for both RAG and long-context understanding, can provide evidence supporting the superiority of APE.
>
> ---
>
> ## **Q4**: And this paper lacks sufficient component analysis and validation.
>
> **A4**:  To further validate the effectiveness of each component in APE, we expanded our ablation studies beyond the three ICL tasks presented in Table 3 of the original paper. Our additional analysis included 8 subsets from LongBench and 5 subsets from ChatRAGBench. The results in Table R7 demonstrate incremental improvements with each component: incorporating shared prefix enhanced performance by 1.75%, adding scaling factor yielded a 0.63% improvement, and introducing attention temperature provided an additional 0.5% gain. Similarly, Table R8 shows consistent improvements from each component, with gains of 2.3%, 0.29%, and 0.79% respectively. These findings further validate the value of each component in APE. Additionally, the minor performance gap between APE and sequential encoding in Table R5, combined with these detailed results, shows that the challenges identified in Section 3.2 in the main paper have been addressed by these three components.
>
> Table R7: Ablation study for each component in APE on  8 subsets from LongBench.
>
> |  Method                   | musique | qasper | 2wikimqa | dureader | hotpotqa | narrativeqa | multifieldqa_zh | multifieldqa_en | Average |
> |---------------------|---------|---------|-----------|-----------|-----------|--------------|-----------------|-----------------|---------|
> | PCW               | 18.82 | 42.59 | 40.99 | 21.57 | 47.09 | 23.29 | 54.40 | 45.05 | 36.73 |
> | + shared prefix | 24.13 | 42.32 | 43.09 | 22.73 | 49.71 | 26.93 | 54.26 | 44.65 | 38.48 |
> | + scaling factor | 26.17 | 41.19 | 44.43 | 23.13 | 47.79 | 30.71 | 54.07 | 45.41 | 39.11 |
> | + attention temperature (APE) | 26.19 | 42.32 | 44.43 | 23.13 | 49.71 | 30.71 | 55.03 | 45.41 | 39.61 |
>
> Table R8: Ablation study for each component in APE on  5 subsets from ChatRAGBench.
>
> | Method                        | INSCIT | Doc2Dial | TopicCQA | Qrecc | QuAC  | Average |
> | ----------------------------- | ------ | -------- | -------- | ----- | ----- | ------- |
> | PCW                           | 19.88  | 32.33    | 28.55    | 46.28 | 31.78 | 31.76   |
> | + shared prefix               | 22.9   | 34.62    | 30.67    | 48.11 | 34.02 | 34.06   |
> | + scaling factor              | 23.24  | 34.88    | 30.67    | 48.32 | 34.65 | 34.35   |
> | + attention temperature (APE) | 23.84  | 34.93    | 33.8     | 48.7  | 34.92 | 35.24   |
>
> ---
>
> **References**:
>
> [1] Bai, Yushi, et al. "Longbench: A bilingual, multitask benchmark for long context understanding." arXiv preprint arXiv:2308.14508 (2023).
>
> [2] Izacard, Gautier, et al. "Unsupervised dense information retrieval with contrastive learning." arXiv preprint arXiv:2112.09118 (2021).
>
> [3] Liu, Zihan, et al. "Chatqa: Surpassing gpt-4 on conversational qa and rag." The Thirty-eighth Annual Conference on Neural Information Processing Systems. 2024.
>
> [4] Li, Zehan, et al. "Towards general text embeddings with multi-stage contrastive learning." arXiv preprint arXiv:2308.03281 (2023).
>
> [5] Pan, Zhuoshi, et al. "Llmlingua-2: Data distillation for efficient and faithful task-agnostic prompt compression." arXiv preprint arXiv:2403.12968 (2024).
>
> [6] Xiao, Guangxuan, et al. "Efficient streaming language models with attention sinks." arXiv preprint arXiv:2309.17453 (2023).
>
> [7] Jin, Hongye, et al. "Llm maybe longlm: Self-extend llm context window without tuning." arXiv preprint arXiv:2401.01325 (2024).

---

> > ### Comment · Reviewer_Yenp · 2024-12-03
> >
> > Thank you for your response, even though it arrived a bit late. After carefully reviewing the additional experiments provided by the authors and considering the comments from other reviewers, I no longer have any outstanding concerns. I believe this work will be beneficial to the community. Therefore, I have decided to increase my score to 8.

---

> > > ### Author Response · Authors · 2024-12-03
> > >
> > > Dear Reviewer Yenp,
> > >
> > > Thanks for reading our response and raising the score! We are happy that our new experiments addressing most of your concerns, and we would include them in our next revision. Thanks again for your time!
> > >
> > > Thanks,
> > >
> > > The Authors

---

### Official Review · Reviewer_BcRN · 2024-11-03

**Soundness:** 3
**Presentation:** 4
**Contribution:** 3
**Rating:** 6
**Confidence:** 3

**Summary:**

The papers introduce a simple and effective modification to parallel decoding which can be used a training-free drop-in method to substantially increase efficiency of inference while retaining most of its accuracy.

**Strengths:**

* The paper evaluates across many different models of different architectures.
* The method is training-free which enables it be easily tested across many model -- assuming a working implementation.
* Shows significant gains on efficiency and would have an real-world impact on deployed models. For example, enabling Pre-caching contexts.

**Weaknesses:**

* Paper does not provide implementation of the method -- this can make it easier for reproducibility. Would it be possible to provide that?
* The paper fixes a windows size but shows no empirical or theoretical analysis of choosing different sizes of this window: either from an efficiency or quality perspective. It would be great to have empirical analysis.
* No complexity analysis written out. This could be particularly relevant if we want to vary window above.

**Questions:**

* Would it be possible to include RULER as an evaluation for long-context task. It has become more standard for LC. Showing the impact of APE on needle in a haystack would be interesting as well.
* Are the tasks where APE fails. For example, Does APE still hold when used for Humaneval or RepoBench?
* What would be the impact of finetuning with examples that had APE applied to then. Would it stop degradation?
* Can you provide an analysis of the impact of changing the "window" size you use for parallel decoding.

---

> ### Author Response · Authors · 2024-12-03
> **Response to Reviewer BcRN**
>
> Thank you for reviewing our paper and for your valuable feedback. Below, we address your concerns point by point and we will revise our paper according to your suggestions. We would appreciate it if you could let us know whether your concerns are addressed by our response.
>
> ---
>
> ## **Q1**: Paper does not provide implementation of the method -- this can make it easier for reproducibility. Would it be possible to provide that?
>
> **A1**: Our PyTorch implementation of APE is available as a proof of concept at https://anonymous.4open.science/r/APE_Rebuttal-93EB. We are currently developing an efficient implementation based on Flash-attention [1].
>
> ---
>
> ## **Q2**: The paper fixes a windows size but shows no empirical or theoretical analysis of choosing different sizes of this window: either from an efficiency or quality perspective. It would be great to have empirical analysis. & Can you provide an analysis of the impact of changing the "window" size you use for parallel decoding.
>
> **A2**:  APE supports dynamic window sizes that adapt to different contexts. The maximum window size is determined by subtracting the combined length of the prefix, query, and maximum generation length from the model's maximum length. We set the window size to its maximum to support longer contexts.
>
> Additionally, we conducted experiments to evaluate the impact of window size on model performance. Using the same long-context inputs, we divided the content into chunks with varying window sizes. The results in Table R1 demonstrate that larger window sizes lead to improved accuracies.
>
> Table R1: Ablation study of window size in APE on the LongBench dataset.
>
> | Window Size | MuSiQue | NarrativeQA | HotpotQA | 2WikiMQA |
> | :-----------: | :---- | :---- | :---- | ----- |
> |     4K    | 24.23 | 28.85 | 47.09 | 39.87 |
> |  8K   | 26.19 | 30.71 | 49.71 | 44.43 |
>
>
> ---
>
> ## **Q3**: No complexity analysis written out. This could be particularly relevant if we want to vary window above.
>
> **A3**: Thank you for your suggestion. We add analysis for the computational complexity of APE and Sequential Encoding. Following Section 2.1 and let the context-augmented input to be
>
> $$\mathcal{S} = {\underbrace{s_{C_0, 0}, \ldots, s_{C_0, l_0}}\_{\text{Context}}, \underbrace{s_{C_1, 0}, \ldots, s_{C_1, l_1}}\_{\text{Context 1}}, \ldots, \underbrace{s_{C_{N-1}, 0}, \ldots, s_{C_{N-1}, l_{N-1}}}\_{\text{Context N-1}}, \underbrace{s_0, \ldots, s_l}\_{\text{Query}}}, $$ where $l_0 + \ldots + l_{N-1} \gg l$, we analyze the prefilling time of Sequential Encoding and APE.
>
> * Sequential Encoding: $O((l_0+...+l_{N-1}+l)^2)$
>
> * APE with pre-caching: $O((l_0+...+l_{N-1}+l)\cdot l)$
>
> * APE without pre-caching: $O(\max(l_0^2, ..., l_{N-1}^2)+((l_0+...+l_{N-1}+l)\cdot l$
>
> Therefore, APE achieves linear complexity with respect to total context length when using pre-caching, resulting in significantly improved efficiency. Even without pre-caching, APE maintains an advantage by allowing independent prefilling of each context. In this case, its complexity scales quadratically with the maximum context length, which is more efficient than Sequential Encoding's quadratic scaling with total context length.
>
> ---
>
> ## **Q4**: Would it be possible to include RULER as an evaluation for long-context task. It has become more standard for LC. Showing the impact of APE on needle in a haystack would be interesting as well.
>
> **A4**: First, we need to clarify that APE is primarily designed for context-augmented generation despite its potential in processing long single contexts as featured in RULER [2] and Needle In A Haystack [3] benchmarks. Here, we evaluated the LLaMA-3.1-8B-Instruct model on these benchmarks using 128K context lengths, with the context divided into 8 chunks of 16K positions each. We also include Quest [4] and ShadowKV [5] as two baselines using dynamic sparse attention and 128K positions. As shown in Table R2, APE demonstrates only a 5% performance decrease compared to Sequential Encoding. This decline is primarily observed in tasks requiring inter-context dependencies, specifically in VT and FWE subtasks. Compared to other baselines, APE even outperforms Quest and approaches ShadowKV despite using only 1/8 positions. These new results further show APE's superiority in long-context understanding.
>
> Table R2: Experiment results of APE on the RULER benchmark.
>
> | Model | N-S1 | N-S2 | N-MK1 | N-MK2 | N-MQ | N-MV | QA-1 | QA-2 | VT | FWE | Average |
> |---|---|---|---|---|---|---|---|---|---|---|---|
> | LLaMA-3.1-8B-Instruct | 100 | 100 | 98.96 | 91.67 | 98.96 | 95.31 | 82.29 | 47.92 | 68.96 | 71.18 | 85.525 |
> | Quest | 100 | 98.96 | 97.92 | 34.38 | 93.49 | 88.54 | 70.83 | 44.79 | 65.63 | 68.4 | 76.294 |
> | ShadowKV | 100 | 100 | 100 | 83.33 | 97.92 | 92.19 | 81.25 | 48.96 | 67.08 | 64.93 | 83.566 |
> | APE | 100 | 96.88 | 97.92 | 85.42 | 97.92 | 93.75 | 75 | 46.88 | 56.04 | 59.72 | 80.953 |

---

> ### Author Response · Authors · 2024-12-03
> **Response to Reviewer BcRN (2/2)**
>
> In Table R3, we further validate that APE can always retrieve the needle from the 128K context on the Needle In A Haystack benchmark.
>
> Table R3: Experiment results of APE on the Needle In A Haystack benchmark.
>
> |                      | 1K   | 10K  | 19K  | 28K  | 37K  | 46K  | 55K  | 64K  | 73K  | 82K  | 91K  | 100K | 109K | 118K | 128K |
> | -------------------- | ---- | ---- | ---- | ---- | ---- | ---- | ---- | ---- | ---- | ---- | ---- | ---- | ---- | ---- | ---- |
> | Average Accuracy (%) | 100  | 100  | 100  | 100  | 100  | 100  | 100  | 100  | 100  | 100  | 100  | 100  | 100  | 100  | 100  |
>
> ---
>
> ## **Q5**: Are the tasks where APE fails.
>
> **A5:** APE fails in scenarios that requires the dependency among contexts, ,particularly when long contexts need to be split into multiple chunks. This limitation is especially evident in tasks requiring long-range dependency. Our experimental results from the VT and FWE subsets in Table R2 demonstrate significant performance degradation in such scenarios.
>
> ---
>
> ## **Q6**: Does APE still hold when used for Humaneval or RepoBench?
>
> **A6:** Yes, APE still hold for several code generation tasks as showcased in Table R4. The results on LCC and RepoBench-P show that APE can outperforms other long-context methods on code generation.
>
> Table R4: Comparison of APE and other methods on long-context code generation.
>
> |  | LCC       | RepoBench-P |
> | :------------ | ----- | ----- |
> | LLaMA-3-8B-Instruct    | 53.22 | 38.15 |
> |    + LLMLingua2     | 16.41     | 20.56       |
> |   + StreamingLLM    | 40.02     | 26.16       |
> |  + Long-context FT  | 55.12     | 43.05       |
> |    + Self-Extend     | 58.01     | 41.83 |
> | + APE | **66.09** | **49.43** |
>
>
> ---
>
> ## **Q7**: What would be the impact of finetuning with examples that had APE applied to then. Would it stop degradation?
>
> **A7:** The model's performance should be improved after fine-tuning for both sequential encoding and APE, as the model encodes the knowledge in examples into its parameters. However, this will lead to significant computational overhead during inference if we fine-tune the model on-the-fly using the retrieved examples, making it not practical.
>
> ---
>
> **References**:
>
> [1] Dao, Tri, et al. "Flashattention: Fast and memory-efficient exact attention with io-awareness." Advances in Neural Information Processing Systems 35 (2022): 16344-16359.
>
> [2] Hsieh, Cheng-Ping, et al. "RULER: What's the Real Context Size of Your Long-Context Language Models?." arXiv preprint arXiv:2404.06654 (2024).
>
> [3] Greg Kamradt. Needle in a haystack - pressure testing llms. 2023.
>
> [4] Tang, Jiaming, et al. "Quest: Query-Aware Sparsity for Efficient Long-Context LLM Inference." arXiv preprint arXiv:2406.10774 (2024).
>
> [5] Sun, Hanshi, et al. "ShadowKV: KV Cache in Shadows for High-Throughput Long-Context LLM Inference." arXiv preprint arXiv:2410.21465 (2024).

---

> > ### Author Response · Authors · 2024-12-04
> > **Additional experiments on the 128K and 1M context-length LOFT benchmark**
> >
> > Dear Reviewer BcRN,
> >
> > Based on the suggestions from Reviewer kKET, we further evaluated our approach on four text retrieval tasks from the LOFT benchmark to represent long-context RAG scenarios, which further validate that APE's scalability in processing hundreds/thousands of retrieved texts with the total length of 1M in parallel. We hope these results, combined to the results on the RULER and Needle-In-A-Haystack datasets, can comprehensively show the effectiveness of APE in extreme long-context scenarios.
> >
> > Here, we use LLaMA-3.1-8B-Instruct as our base model, with performance measured by the F1 score between ground truth and model predictions. To simulate realistic scenarios where RAG systems may retrieve suboptimal content, we randomly sampled passages from the corpus as context, enabling analysis of how performance scales with additional context integration (Otherwise, we only need a short-context LLM to achieve a good performance if the retriever is very accurate, which is not the setting that we target on). For the 128k version of LOFT, we compared APE against Sequential Encoding using varying context lengths in Table R9.
> >
> > Table R9. Comparison of APE and Sequential Encoding on four text retrieval tasks from the 128k version of LOFT benchmark using the LLaMA-3.1-8B-Instruct model.
> >
> > |                   | Inference Time | Arguana | Fever | NQ    | SciFact | Average |
> > | :------------------------------ | :-------- | :--------- | :--------- | :---------- | :---------- | ------------------------------- |
> > | 8K, Sequential | 847            | 11.15   | 7.78 | 17.78 | 7.74 | 11.11 |
> > | 16K, Sequential | 1277           | 11.20   | 9.78 | 17.81 | 9.49 | 12.07 |
> > | 32K, Sequential | 2413 | 12.15 | 10.22 | 15.78 | 9.74 | 11.97 |
> > | 64K, Sequential | 5870 | 15.34 | 13.12 | 19.64 | 16.12 | 16.06 |
> > | 128K, Sequential | 18039 | 12.84 | 14.19 | 24.54 | 16.88 | 17.11 |
> > | 128K, APE | 1825 | 16.32 | 14.70 | 21.91 | 15.72 | 17.16 |
> >
> > Our analysis yielded several key findings:
> >
> > * APE demonstrates superior performance on average compared to all baselines, validating APE’s scalability in processing hundreds/thousands of retrieved texts in parallel.
> >
> > * APE achieves performance comparable to the 128K Sequential Encoding baseline while utilizing the same context. Although transferring from sequential encoding to APE would typically be expected to cause performance degradation, we hypothesize that APE recovers this by effectively shortening the distance between the query and all examples. This mechanism helps address the “lost in the middle” problem common in long-context LLMs.
> >
> > * APE demonstrates remarkable computational efficiency, with its 128K variant requiring inference times comparable to Sequential Encoding at much shorter context lengths (between 16K and 32K), while achieving approximately 10 times faster processing compared to Sequential Encoding at equivalent context length.
> >
> > We also compared APE with Sequential Encoding using the LLaMA-3.1-8B-Instruct model on the 1m version of LOFT, as shown in Table R10. While APE improves 5.26% on average using 8$\times$ more retrieved texts, it only requires 1/3 inference time by pre-caching these texts.
> >
> > Table R10. Comparison of APE and Sequential Encoding on four text retrieval tasks from the 1m version of LOFT benchmark using the LLaMA-3.1-8B-Instruct model.
> >
> > |                   | Inference Time | Arguana | Fever | NQ    | SciFact | Average |
> > | :------------------------------ | :-------- | :--------- | :--------- | :---------- | :---------- | ------------------------------- |
> > | 128K, Sequential | 18039          | 11.28 | 8.13 | 17.55 | 7.54 | 11.13 |
> > | 1M, APE          | 5073           | 15.89 | 15.13 | 20.78 | 14.55 | 16.59 |

---

### Official Review · Reviewer_YT3n · 2024-11-07

**Soundness:** 3
**Presentation:** 3
**Contribution:** 2
**Rating:** 6
**Confidence:** 2

**Summary:**

The paper presents Adaptive Parallel Encoding (APE) in order to improve efficiency and performance in language models handling RAG and ICL tasks. APE overcomes limitations in sequential encoding, which requires re-encoding context data and suffers from context window restrictions, by employing parallel encoding with adaptive adjustments. These changes include a shared prefix, scaling factors, and a modified attention temperature, aligning parallel encoding closely with sequential encoding. Experiments show APE achieves a high speedup in long contexts and outperforms slightly other methods in accuracy for long contexts.

**Strengths:**

- The method sounds intuitive. APE extends the usable context length for language models, overcoming limitations in the context window and enabling efficient handling of much larger inputs without additional training.
- The performance is great esp. in latency
- The analysis in Section 3 looks quite novel and interesting

**Weaknesses:**

- The performance seems quite sensitive on specific parameters

**Questions:**

- Will we conclude anything between the task and the choice of hyperparameters?
- The latency improvement in Table 2 would be more interesting if other baselines are included.

---

> ### Author Response · Authors · 2024-12-03
> **Response to Reviewer YT3n**
>
> Thank you so much for the insightful and valuable comments! They are very helpful for further improving the quality of our paper. We will revise our manuscript in the next version to address all of your concerns.
>
> ---
>
> ## **Q1**: The performance seems quite sensitive on specific parameters.
>
> **A1**: Our APE, despite introducing three hyperparameters (shared prefix, scaling factor, and attention temperature), is not very senstive on these hyperparameters. First, the shared prefix is only applied to tasks lacking existing prefixes (e.g., GSM8K and TriviaQA). Our experimental results in Table R1 demonstrate that introducing this shared prefix increases performance by 7.38% on average, while variations in prefix length only result in a 0.4% difference in performance. These findings indicate that while the presence of a shared prefix is beneficial, APE's performance is not significantly sensitive to the prefix length.
>
> Table R1: Ablation study of different prefix lengths on GSM8K and TriviaQA using the LLaMA-3-8B-Instruct model.
>
> | Prefix Length |    0  | 10    | 20    | 40    |
> | :-----------: | ----: | ----: | ----: | ----: |
> |   GSM8K     | 62.17 | 72.10 | 71.87 | 71.69 |
> |   TriviaQA    | 67.48 | 72.31 | 72.51 | 72.65 |
>
> For the scaling factor $s$ and attention temperature $T$, we evaluated various combinations across four QA tasks from the LongBench dataset, as shown in Table R2. While these hyperparameters are relatively sensitive compared to the prefix length, their optimal values fall within a narrow range. In our experiments, both $s$ and $T$ were selected from {0.7, 0.8, 0.9, 1.0}, making hyperparameter searching efficient on the validation set. Notably, despite examples of each task on LongBench containing varying numbers of contexts, APE's hyperparameters demonstrated generalization ability across examples with different context lengths, indicating low sensitivity.
>
> Table R2: Ablation study of  the scaling factor s and attention temperature T on four subsets from LongBench using the LLaMA-3-8B-Instruct model.
>
> | ($s$, $T$) | MuSiQue | NarrativeQA | HotpotQA | 2WikiMQA |
> | :-----------: | :---- | :---- | :---- | ----- |
> | (1.0, 1.0) | 24.13   | 26.93       | 49.71    | 43.09    |
> | (1.0, 0.9) | 26.17   | 30.71       | 47.79    | 41.19    |
> | (0.9, 0.9) | 24.32   | 27.93       | 49.6     | 38.21    |
> | (0.8, 0.8) | 26.19   | 27.01       | 47.32    | 35.41    |
>
> ---
>
> ## **Q2**: Will we conclude anything between the task and the choice of hyperparameters?
>
> **A2**:  In different tasks, we may incorporate different number of retrieved texts from external sources. With this number increase, we observe a consistent trend that both the scaling factor s and attention temperature T decrease, aligning with our observations in Section 3.2.
>
> ---
>
> ## **Q3**: The latency improvement in Table 2 would be more interesting if other baselines are included.
>
> **A3**: Thank you for your suggestion. We additionally add MInference [1] as another baseline for prefilling acceleration. As shown in Table R3, APE achieves up to **28$\times$** acceleration in prefilling compared to Sequential Encoding with MInference for 128K-token contexts. Furthermore, APE's prefilling overhead scales linearly, consuming less than 10% of the total inference time, while baseline methods require over 50% as context length increases. APE also demonstrates superior versatility across various settings, whereas MInference introduces additional overhead and slower inference speeds when processing short contexts and large batches.
>
> Table R3: Latency on H100 GPU: [prefilling / total time (ms)]. We compared different strategies with various context lengths and batch sizes.
>
> |       Context Length            | 4K        | 8K        | 16K        | 32K        | 64K         | 128K        |
> | :------------------------------ | :-------- | :-------- | :--------- | :--------- | :---------- | :---------- |
> |       Sequential (bsz=1)        | 132/2383  | 276/2558  | 635/3204   | 1646/4713  | 4892/8804   | 16635/22252 |
> | Sequential + MInference (bsz=1) | 964/3221  | 1297/3609 | 1623/4201  | 2206/5274  | 3563/7458   | 6457/12085  |
> |        APE (bsz=1)         | 28/2279   | 40/2322   | 68/2637    | 122/3189   | 226/4138    | 421/6038    |
> |       Sequential (bsz=4)        | 523/3129  | 1084/4179 | 2558/6474  | 6599/12055 | 19468/28367 | 65928/83869 |
> | Sequential + MInference (bsz=4) | 3789/6373 | 4995/8099 | 6300/10244 | 8470/13944 | 13462/22313 | 24199/42013 |
> |        APE (bsz=4)         | 69/2675   | 94/3189   | 150/4066   | 298/5754   | 430/9329    | 858/18799   |
>
>
> ---
>
> **References**:
>
> [1] Jiang, Huiqiang, et al. "Minference 1.0: Accelerating pre-filling for long-context llms via dynamic sparse attention." arXiv preprint arXiv:2407.02490 (2024).

---

### Author Response · Authors · 2024-12-03
**Response to All Reviewers (1/2)**

We sincerely apologize to all reviewers and Area Chairs for submitting our rebuttal in the final hours of the discussion period. We conducted extensive additional experiments that took longer than anticipated, consuming most of the available time. We would be grateful if you could still review our rebuttal during the discussion/decision period.

We thank reviewers [R1(YT3n), R2(BcRN), R3(Yenp), R4(onHd), R5(kKET)] for their thoughtful and highly supportive feedback! We were glad that the reviewers found the problem **practical and significant [R2, R3]**, the observations and analyses **insightful and thorough [R3, R4]**, the method **training-free and deployable[R2, R4]**, the experimental results **comprehensive and impressive [R4]**.

While several reviewers [R2, R3, R4, R5] focus on APE's potential for long-context LLMs, and R3 requests additional experiments for RAG and multi-document QA, we would like to first clarify APE's primary contribution. **Our method focuses on context-augmented generation involving multiple predetermined contexts from an external dataset, rather than processing single long-contexts in real-time. Therefore, the unique advantage of APE is its ability to pre-cache these contexts, significantly reducing prefilling time**, as shown in Table 2 in the original paper. With the prevalence of context-augmented generation  in real-world deployment, we hope that APE can motivate the development of pre-cacheable strategies in practice. Therefore, APE is completely different from the fixed step sparse attention mentioned by R5 and other length extension methods, which require prefilling the contexts on-the-fly in context-augmented generation settings.

To comprehensively address the reviewers' concerns while keeping the original focus, we have expanded our evaluation on both sides, with the detailed results provided in responses to each reviewer:

* [R3] RAG on the LongBench dataset [1]. In this setting, we preprocess long contexts into chunks and utilize Contriever [2] to retrieve the top-k most relevant chunks on LongBench. On average, APE leads to 5.6% and 4.5% improvement compared to Sequential Encoding and PCW.  These findings suggest that APE enhances RAG performance by enabling the retrieval of longer, more comprehensive contexts without compromising inference speed, as pre-cached contexts can be loaded directly.

* [R2, R4, R5] RAG on the ChatRAGBench dataset [3]. In this setting, we use three different retrievers to retrieve the top-5 relevant contexts on the ChatRAGBench dataset. The results demonstrate that APE results in a no more than 1% performance degradation compared to sequential encoding using the same contexts, which we consider acceptable. Furthermore, when directly processing all contexts, APE outperforms all sequential encoding baseline.

* [R2, R4, R5] Long-context understanding on the RULER dataset [4]. In this setting, APE demonstrates only a 5% performance decrease compared to sequential encoding for 128k context length. This decline is primarily observed in tasks requiring inter-context dependencies, specifically in VT and FWE subtasks. Compared to other baselines, APE even outperforms Quest [7] and approaches ShadowKV [8] despite using only 1/8 positions. This indicates that APE can scale to longer contexts.

* [R2, R4, R5] Long-context understanding on the Needle in a haystack dataset [5]. In this setting, APE can always retrieve the needle from the 128k-length context on the Needle In A Haystack dataset.

* [R2, R3, R4, R5] Long-context understanding on the LongBench dataset [1]. In this setting, APE demonstrates superior long context understanding, outperforming LLMLingua2 [9], StreamingLLM [10], Long-context Fine-tuning, and Self-Extend [11] by an average improvement of 2.15% across 11 subtasks compared to the SOTA baseline.

We hope that these new results, combined with the efficiency analysis, can (i) further support the novelty of APE in context-augmented generation as the major contribution, (2) provide more evidence for the effectiveness of APE in long-context understanding as a by-product.

---

> ### Author Response · Authors · 2024-12-03
> **Response to All Reviewers (2/2)**
>
> **References**:
>
> [1] Bai, Yushi, et al. "Longbench: A bilingual, multitask benchmark for long context understanding." arXiv preprint arXiv:2308.14508 (2023).
>
> [2] Izacard, Gautier, et al. "Unsupervised dense information retrieval with contrastive learning." arXiv preprint arXiv:2112.09118 (2021).
>
> [3] Liu, Zihan, et al. "Chatqa: Surpassing gpt-4 on conversational qa and rag." The Thirty-eighth Annual Conference on Neural Information Processing Systems. 2024.
>
> [4] Li, Zehan, et al. "Towards general text embeddings with multi-stage contrastive learning." arXiv preprint arXiv:2308.03281 (2023).
>
> [5] Hsieh, Cheng-Ping, et al. "RULER: What's the Real Context Size of Your Long-Context Language Models?." arXiv preprint arXiv:2404.06654 (2024).
>
> [6] Greg Kamradt. Needle in a haystack - pressure testing llms. 2023.
>
> [7] Tang, Jiaming, et al. "Quest: Query-Aware Sparsity for Efficient Long-Context LLM Inference." arXiv preprint arXiv:2406.10774 (2024).
>
> [8] Sun, Hanshi, et al. "ShadowKV: KV Cache in Shadows for High-Throughput Long-Context LLM Inference." arXiv preprint arXiv:2410.21465 (2024).
>
> [9] Pan, Zhuoshi, et al. "Llmlingua-2: Data distillation for efficient and faithful task-agnostic prompt compression." arXiv preprint arXiv:2403.12968 (2024).
>
> [10] Xiao, Guangxuan, et al. "Efficient streaming language models with attention sinks." arXiv preprint arXiv:2309.17453 (2023).
>
> [11] Jin, Hongye, et al. "Llm maybe longlm: Self-extend llm context window without tuning." arXiv preprint arXiv:2401.01325 (2024).

---

### Author Response · Authors · 2024-12-03
**Pytorch Implementation of APE**

Our PyTorch implementation of APE is available as a proof of concept at https://anonymous.4open.science/r/APE_Rebuttal-93EB.

---

### Meta-Review · Area_Chair_2n88 · 2024-12-20

**Metareview:**

The paper introduces Adaptive Parallel Encoding (APE), a novel method designed to enhance efficiency and performance in language models when handling multiple external contexts, such as Retrieval-Augmented Generation (RAG) and In-Context Learning (ICL). APE achieves this by pre-caching key-value states of contexts separately and enabling position reuse during inference. The method incorporates three key components: 1) A shared prefix to align initial token distributions, 2) A scaling factor to offset the increase in attention weights, and 3) A lower attention temperature to prioritize semantically important tokens. The authors demonstrate that APE achieves an impressive 976× speedup for long-context generation while maintaining 93% accuracy compared to sequential encoding.

Overall, the reviewers unanimously acknowledge the soundness and significance of the proposed techniques, and the substantial efficiency improvements. Given the potential impact of APE in reducing inference costs for large models in RAG and ICL tasks, we recommend accepting this submission. But for authors, please follow the reviewers' feedback and your promise to revise the paper.

**Additional Comments On Reviewer Discussion:**

Here I mainly list the main concerns.

1)	The performance seems quite sensitive to specific parameters. (Reviewers YT3n, onHd)
  The authors add extra experiments to show that the three hyperparameters (shared prefix, scaling factor, and attention temperature), is not very sensitive to these hyperparameters.

2)	Latency improvement in Table 2 would be more interesting if other baselines are included. (Reviewer YT3n)
  The authors add MInference as another baseline for prefilling acceleration. As shown in Table R3, APE achieves up to 28× acceleration in prefilling compared to Sequential Encoding with MInference for 128K-token contexts.

3)	Impact of changing the "window" size you use for parallel decoding, and complexity change (reviewer BcRN)
The authors explain that 1) APE supports dynamic window sizes that adapt to different contexts; and 2) adding more experiments to show the performance of different window sizes.

4)	The 128K and 1M context-length LOFT benchmark (reviewer BcRN)
The authors add extra experiments to show the effectiveness of APE under 128k and 1M context.

5)	The narrow scope of evaluation fails to provide comprehensive evidence of the method's effectiveness. ( Reviewers Yenp, onHd)
 The authors add many extra experiments to show the effectiveness of APE.

These concerns are well resolved in my view.

---

### Decision · Program_Chairs · 2025-01-22

Accept (Poster)